# FedCova: Robust Federated Covariance Learning Against Noisy Labels

## Abstract

This paper addresses the critical challenge of federated learning (FL) under noisy labels by exploiting intrinsic robustness grounded in covariance structures. Noisy labels in distributed datasets induce severe local overfitting and consequently compromise the global model in FL. Most existing solutions rely on selecting clean devices or aligning with public clean datasets, rather than endowing the model itself with robustness. In this paper, we propose *FedCova*, a noise-resistant federated covariance learning framework, to enhance the model's intrinsic robustness via a new perspective on feature covariances. Specifically, FedCova encodes data into a discriminative but resilient feature space to tolerate label noise. Built upon mutual information maximization, we design a novel objective for federated lossy feature encoding, which is driven solely by the feature covariances of different classes with an error tolerance term. Leveraging feature subspaces characterized by covariances, we construct a subspace-augmented federated classifier. FedCova unifies three key processes through the covariance: (1) training the network for feature encoding, (2) constructing a classifier directly from the learned features, and (3) correcting labels based on feature subspaces. The server aligns the federated classifier via covariance aggregation, which devices use to build local external correctors for relabeling, avoiding self-correction. We implement FedCova under heterogeneous data distribution across various noisy settings. Experimental results on CIFAR-10/100 and real-world noisy dataset Clothing1M demonstrate the superior robustness of FedCova compared with the state-of-the-art methods.

## 1 Introduction

Exploring robustness to noisy labels is critical in machine learning, as it enhances the adaptability of large-scale model training to massive, coarsely labeled datasets Chen et al. (2019). Traditional approaches primarily refine the training process with several *offline* noise-resistant strategies, typically detecting, selecting, or correcting noisy samples Cheng et al. (2022); Ren et al. (2018); Englesson & Azizpour (2021); Xia et al. (2024). These heuristic methods often exhibit fragile robustness, particularly under severe label noise. To address this, many recent algorithms broaden the learning framework by leveraging auxiliary clean datasets or building multiple model structures Chen et al. (2019); Zheng et al. (2024). While such extensions can improve performance, they remain dependent on *additional resources* rather than on strengthening the inherent robustness of models via a deeper understanding of the data.

Within this broader context, federated learning (FL) under noisy labels emerges as a further challenging scenario. As intelligence tending to terminals, it is increasingly expected to be ubiquitous yet localized and interconnected yet privacy-preserving Tan et al. (2023); Chen et al. (2024). As a typical distributed paradigm, FL facilitates collaborative learning over edge devices, preserving privacy through data locality McMahan et al. (2017); Kairouz et al. (2021). However, the distributed training nature makes FL overreliant on limited and localized data Kairouz et al. (2021); Li et al. (2019). On the one hand, from the perspective of data sources, labels collected on edge devices are vulnerable to annotation errors, sensor faults, and adversarial label attacks Song & et al. (2022); Han et al. (2020). On the other hand, local models are more fragile to limited noisy data, inducing misguided overfitting and subsequently contaminating the global model after aggregation. Following model broadcasts further propagate noisy updates, accumulating over iterations. Meanwhile, most existing FL studies have concentrated on heterogeneity mitigation, further intensifying the reliance on accurate data

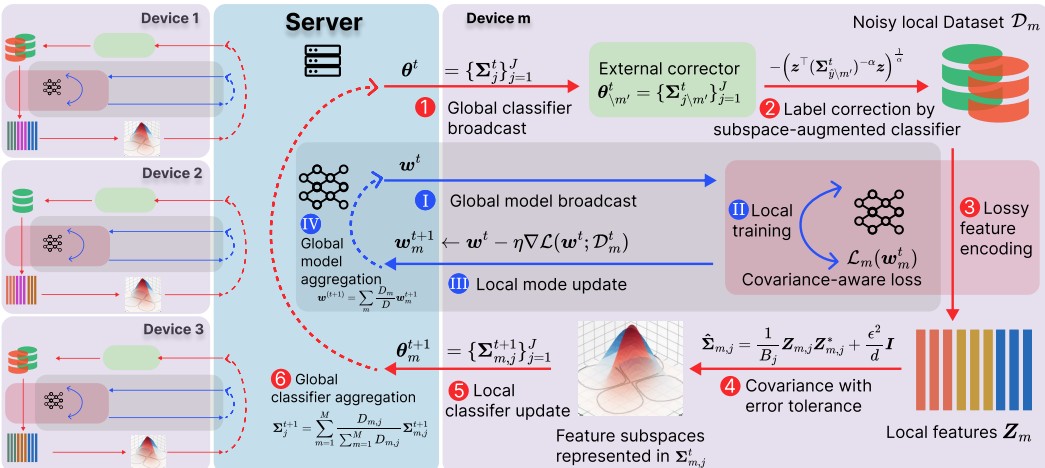

Figure 1: Overview of the FedCova framework. The green data is clean, while the red data is mislabeled. The pink rectangle indicates covariance-aware feature learning. The gray rectangle with the circle of blue arrows ❶-❹ indicates the conventional FL processes, around which the circle of red arrows ❶-❻ in FedCova constructs a fortress to guard against label noise **under the flow of the covariances** $\Sigma$. Specifically, the server first broadcasts ❶ the global model and ❶ global classifier to edge devices. ❷ Label correction can be conducted then, after which devices ❷ ❸ perform local feature learning and update ❸ the local models and ❹ ❺ local classifiers to the server. The latter then ❹ ❻ aggregates them for the next rounds of iterations.

Karimireddy et al. (2020); Zhang et al. (2022). These methods suffer potential deterioration once noisy labels are introduced, since overfitting to noisy data undermines their distributional alignment strategies Arpit et al. (2017); Zhang et al. (2021); Lu et al. (2024); Xu et al. (2022).

Recent efforts toward FL robustness under noisy labels have primarily relied on transferring offline techniques from centralized learning, including sample selection and robust loss regularization Han et al. (2018); Song & et al. (2022); Yi & Wu (2019), into FL frameworks Di et al. (2024); Xu et al. (2022); Wu et al. (2023a). However, these methods largely ignore FL's unique properties, including its distributed topology for inherent cross-validation Ovi et al. (2023) and the potential of latent representation sharing for capturing distributed structures Tan et al. (2022). In contrast, some works harness FL's distributed nature—for instance, through dual-model co-training Ji et al. (2024) or cross-device distillation Tsouvalas et al. (2024); Lu et al. (2024)—but their effectiveness hinges on extra resources, notably redundant models or clean public datasets. This exposes a fundamental bottleneck: under noisy labels, *models remain tied to strict prediction–label alignment*, often limited by objectives like cross-entropy, which inherently pushes predictions toward potentially incorrect labels. Regarding the correction strategy, most approaches identify corrupted samples via prediction or loss consistency, then depend on clean clients to relabel or reweight during training Tam et al. (2023); Yang et al. (2022); Xu et al. (2022). Yet such reliance on clean datasets is impractical: when clean clients are few—or hold only limited samples or classes under non-independently and identically distributed (non-i.i.d.) data—these methods may collapse, as they do not intrinsically exploit the underlying data characteristics.

Representation learning underscores the importance of extracting discriminative and informative features that capture intrinsic data structures Bengio et al. (2013); Han et al. (2020). Motivated by this, leveraging the feature space in FL with noisy labels can provide a more reliable basis for robust learning, since feature statistics are less distorted by label noise Arpit et al. (2017); Zhang et al. (2023); Chan et al. (2022). Contrastive learning has been applied to centralized learning under noisy labels by capturing relative relationships between sample features Li et al. (2021); Yi et al. (2022). However, it requires contrasting all sample pairs, which is infeasible in FL, where data remain distributed across devices and cross-device comparison is restricted. Consequently, such methods underperform in FL Zhang et al. (2023); Duan et al. (2022). Attempts to leverage features in FL typically aggregate local feature means/centroids with alignment regularization Tan et al. (2022); Xu et al.. But noisy labels directly bias centroids, preserving noise and compounding the overfitting already caused by the CE loss.

To bridge the gaps observed in existing studies, we propose **FedCova**, a robust ***Cova**riance*-**a**ware **Fed**erated feature learning framework designed to mitigate the impact of noisy labels in FL (overview in Fig. 1). Specifically, FedCova treats federated learning as a feature encoder that learns *discriminative yet noise-resilient representations* by capturing the intrinsic statistical structure in features. We exploit the *covariance* of class-conditional features to construct a feature space that preserves class separability while remaining robust to noise. FedCova seamlessly integrates three key processes through the feature *covariance*: (1) network training for feature encoding, (2) intrinsic classifier construction directly from learned features, and (3) feature subspace-based label correction. We introduce an information-theoretic training objective based on mutual information maximization, which depends solely on feature *covariance*, and extend it to a lossy variant with an error tolerance term. Driven by this, the learned encoding decouples the dependencies between the predicted output and the observed label, while capturing discriminative statistics of classes within orthogonal feature subspaces. On the server side, aggregated *covariances* are used to align a global classifier and ensure consistency of feature representations across devices. Based on these global *covariances*, FedCova constructs a maximum a *posterior* (MAP) classifier and further generalizes it via subspace augmentation to strengthen discrimination. Upon receiving the global classifier, each device leverages it to independently design a local corrector for relabeling, which enables effective label correction while preventing the pitfalls of self-correction. Our main contributions are summarized as follows:

- We propose FedCova, a unified *covariance*-aware federated feature learning framework for robust federated learning against noisy labels. Through feature *covariance*, FedCova integrates feature encoding, intrinsic classifier construction, and subspace-based label correction.

- We introduce a *covariance*-based information-theoretic loss function for federated lossy feature learning. Grounded in maximizing mutual information, the objective constrains the *covariance* structure of class-conditional features and enhances both discrimination and noise resilience under an error tolerance term over it.

- We develop a federated classifier alignment strategy via *covariance* aggregation, and complement it with a local correction strategy. From the intrinsic MAP classifier, we generalize it with a subspace augmentation design. Each client then leverages the global knowledge to construct an external corrector to correct noisy labels while preventing self-bias.

- We conduct comprehensive experiments across a variety of noisy device ratios and sample noise ratios over CIFAR-10/100 and real-world noisy dataset Clothing1M under non-i.i.d. data distribution, demonstrating superior robustness of FedCova against noisy labels compared to state-of-the-art methods.

## 2 PRELIMINARIES

**Federated Feature Learning.** Training with noisy labels undermines performance by corrupting the input–label mapping, leading to overfitting and incorrect predictions. Rather than relying solely on this direct mapping, we aim to explore whether representations in a discriminative feature space can offer greater robustness by mediating the relationships among images, features, and labels. For detailed discussions of related works and previous solution routes, we refer readers to Appendix A.

Consider a federated learning (FL) system comprising $M$ edge devices to collaboratively train a deep learning model. The FL system aims to learn the discriminative and representative feature structure from the high-dimensional multi-class images for the classification task. Denote the data, feature, and label spaces by $\mathcal{X}$, $\mathcal{Z}$, and $\mathcal{Y}$, respectively. Let $\boldsymbol{x} \in \mathcal{X}$, $\boldsymbol{z} \in \mathcal{Z}$, and $y \in \mathcal{Y}$ denote instances of corresponding random variables $X$, $Z$, and $Y$. Consider $\mathcal{Y} = \{1, 2, \ldots, J\}$, where $J$ is the number of classes. Denote by $\mathcal{D}_m = p(\boldsymbol{x}, y|m)$ the dataset distribution of device $m \in \mathcal{M}$, where $\mathcal{M}$ is the device set. The joint distribution of a sample in a device is denoted by $p(\boldsymbol{x}, y, m) = p(m)p(\boldsymbol{x}, y|m) = p(m)p(y|m)p(\boldsymbol{x}|m, y)$. The global mixture distribution of all devices is $\mathcal{D} = \mathbb{E}_{m \sim p(m)}\mathcal{D}_m$. The FL system is trained by minimizing the loss function as

$$\min_{\boldsymbol{w}^t} \quad \mathcal{L}(\boldsymbol{w}^t) := \mathbb{E}_{m \sim p(m)}\mathcal{L}_m(\boldsymbol{w}^t), \ \text{ with } \ \mathcal{L}_m(\boldsymbol{w}^t) := \mathbb{E}_{(\boldsymbol{x},y) \sim \mathcal{D}_m}\ell(f_{\boldsymbol{w}^t}(\boldsymbol{x}), y) \tag{1}$$

where $f_{\boldsymbol{w}^t} : \mathcal{X} \to \mathcal{Z}$ is the network with the global model parameter $\boldsymbol{w}^t$ mapping from the input data to the feature representation in the $t$-th communication round. To be specific, we intend to learn the representation $\boldsymbol{z} \in \mathbb{R}^d$ of a data sample $\boldsymbol{x} \in \mathbb{R}^D$ with a distribution $p_{\boldsymbol{w}^t}(\boldsymbol{z}|\boldsymbol{x})$ parameterized by the

neural network $\boldsymbol{w}^t \in \mathbb{R}^s$. The FL network outputs $\boldsymbol{z} = f_{\boldsymbol{w}^t}(\boldsymbol{x})$ as a deterministic feature encoder, which can be viewed as a particular instance of the probabilistic one by recognizing its equivalent representation as $p_{\boldsymbol{w}^t}(\boldsymbol{z}|\boldsymbol{x}) = \delta(\boldsymbol{z} - f_{\boldsymbol{w}^t}(\boldsymbol{x}))$, where $\delta(\cdot)$ is the Dirac delta function.

In the communication round $t$, each device $m$ performs the local training based on stochastic gradient descent (SGD) over local dataset $\mathcal{D}_m$ from $\boldsymbol{w}^t$ to obtain the new model parameters $\boldsymbol{w}_m^{t+1} \leftarrow \boldsymbol{w}^t - \eta \nabla \mathcal{L}(\boldsymbol{w}^t; \mathcal{D}_m)$, and uploads it to the server. The server aggregates the local updates as $\boldsymbol{w}^{t+1} = \mathbb{E}_{m \sim p(m)} \boldsymbol{w}_m^t$, which are then broadcast to edge devices for the next round of training.

**Priors.** Usually, the geometric and statistical characteristics of the learning features $\boldsymbol{z}$ remain obscure in the latent layer of the neural network. To drive the network to generate informative features for robust representation to resist noisy labels, we model the features as following a certain probabilistic distribution. Specifically, we introduce a Gaussian mixture (GM) prior over the learned features, where each Gaussian component corresponds to a distinct class, i.e.,

$$p(\boldsymbol{z}) = \sum_{j=1}^{J} p(\boldsymbol{z}, y = j) = \sum_{j=1}^{J} p(j) p(\boldsymbol{z} \mid y = j) = \sum_{j=1}^{J} \pi_j^t \mathcal{N}(\boldsymbol{\mu}_j^t, \boldsymbol{\Sigma}_j^t), \tag{2}$$

with the parameters $\boldsymbol{\theta}^t = \{\pi_j^t, \boldsymbol{\mu}_j^t, \boldsymbol{\Sigma}_j^t\}_{j=1}^{J}$, where $\pi_j^t = p(j)$ denotes the component weight of $j$-th class and $\boldsymbol{\mu}_j^t$ and $\boldsymbol{\Sigma}_j^t$ denote the mean and covariance matrix of feature $\boldsymbol{z}$ in the $j$-th class in the $t$-th communication round.

Unlike conventional priors that assign distinct means to each class and impose strong penalties on the class means Tan et al. (2022), we assume all component means are zero, i.e., $\boldsymbol{\mu}_j^t = \boldsymbol{0}, \forall j \in [1, J]$. This zero-mean design is to remove the reliance on explicit class centers, which are intuitively more prone to bias due to label noise. Instead, we aim for the system to model the covariance structure of the features. By eliminating the effect of the mean, we not only reduce the number of parameters to be estimated in the representations but also encourage the model to capture the distinctive patterns and inter-feature dependencies of each class purely through the covariance matrices, thereby preserving the intrinsic structure of the data in the learned features.

# 3 LOSSY LEARNING OBJECTIVE

**Information-Theoretic Loss** We intend to discover a representation $p_{\boldsymbol{w}^t}(\boldsymbol{z} \mid \boldsymbol{x})$ that contains the maximal information of its label $y$. From the information maximization principle Bell & Sejnowski (1995), a typical optimization objective is to maximize the mutual information, which measures the amount of information shared between the two variables, or equivalently, the statistical dependency between them. Based on this, the objective function for each device $m$ is to maximize the mutual information between $Z_m$ and $Y_m$, or equivalently,

$$\min_{\boldsymbol{w}_m^t} \quad \mathcal{L}_m(\boldsymbol{w}_m^t) := -I(Z_m; Y_m) = h(Z_m \mid Y_m) - h(Z_m), \tag{3}$$

where $h(\cdot)$ is the differential entropy function. We approximate the distribution of $Z_m \sim \mathcal{N}(\boldsymbol{0}, \Sigma_m)$ as a Gaussian distribution with $\Sigma_m$ being its covariance matrix. Based on the GM prior assumptions, the negative mutual information loss in Eq. (3) can be expressed as

$$\mathcal{L}_m(\boldsymbol{w}_m^t) = \frac{1}{2} \sum_{j=1}^{J} \pi_{m,j}^t \log \det \left( \boldsymbol{\Sigma}_{m,j}^t \right) - \frac{1}{2} \log \det \boldsymbol{\Sigma}_m^t. \tag{4}$$

Each client is trained using the objective in Eq. (4), which enables the local models $\boldsymbol{w}_m^t$ to learn class-discriminative features. Note that this objective function is independent of the feature means, since the entropy of Gaussian variables is determined exclusively by their covariance matrices. This observation is consistent with our assumption of zero-mean GM priors. It further implies that focusing on the covariance, from the perspective of mutual information maximization, may allow us to learn reasonably effective features.

**Lossy Representation.** The objective in Eq. (4) seeks to maximize the mutual information between the feature $Z_m$ and its label $Y_m$. Yet, when labels are noisy, the coupling from features to noisy labels

still implicitly affects training when calculating the covariance $\mathbf{\Sigma}_{m,j}^t$ for each class, leaving the model susceptible to overfitting. Consider this, we propose a lossy variant of the objective by introducing controlled variability into the feature space through its covariance structure. In the context of FL under noisy labels, let $\mathcal{Y}', Y', y'$ and $\mathcal{Z}', Z', z'$ denote the space, random variable, and instance of the noisy labels and lossy features, respectively. To tolerate noisy labels, we now introduce elasticity in the feature space $\mathcal{Z}'$. Under the supervision of noisy labels $y'$ to encode the feature from $\boldsymbol{x}$, we regard the output $\boldsymbol{z}'$ as a noisy version of the accurate representation $\boldsymbol{z}$. Therefore, we are actually dealing with $f'_{\boldsymbol{w}^t} : \mathcal{X} \to \mathcal{Z}'$ for FL under noisy labels. Without loss of generality, we model the deviation of the encoded feature as an additive Gaussian error as

$$\boldsymbol{z}'_m = \boldsymbol{z}_m + \boldsymbol{n}, \ \ \text{with} \ \ \boldsymbol{n} \sim \mathcal{N}\left(\boldsymbol{0}, \frac{\epsilon^2}{d}\boldsymbol{I}\right), \tag{5}$$

where $\epsilon^2$ is the square error tolerance of the distortion and $\boldsymbol{n}$ is independent of the feature $\boldsymbol{z}$ so that $\mathbb{E}[\|\boldsymbol{z}'_m - \boldsymbol{z}_m\|_2^2] = \epsilon^2$. During training for each client $m$, it is only accessible to the samples of features to estimate the covariance matrices. For a batch of $B$ training data $\{\boldsymbol{x}_m^i, y_m'^i\}_{i=1}^B$, the FL framework encodes the data samples $\boldsymbol{X}_m \triangleq [\boldsymbol{x}_m^1, ..., \boldsymbol{x}_m^B]$ to receive the lossy version of the feature samples $\boldsymbol{Z}_m \triangleq [\boldsymbol{z}_m^1, ..., \boldsymbol{z}_m^B]$. Thus, the estimated covariance matrix of the lossy features is

$$\hat{\mathbf{\Sigma}}_m = \frac{1}{B}\sum_{i=1}^{B} \boldsymbol{z}_m'^i \boldsymbol{z}_m'^{i*} = \frac{1}{B}\boldsymbol{Z}_m \boldsymbol{Z}_m^* + \frac{\epsilon^2}{d}\boldsymbol{I}, \tag{6}$$

where $(\cdot)^*$ is the transpose operation. Similarly, the covariance matrix of each class $\hat{\mathbf{\Sigma}}_{m,j} = \frac{1}{B_j}\boldsymbol{Z}_{m,j}\boldsymbol{Z}_{m,j}^* + \frac{\epsilon^2}{d}\boldsymbol{I}$, where $B_j$ is the sub-batch in $B$ including samples of the $j$-th class and $\boldsymbol{Z}_{m,j} = [\boldsymbol{z}_{m,j}^1, ..., \boldsymbol{z}_{m,j}^{B_j}]$ are the corresponding features samples. Therefore, the loss function in Eq. (4) can be reformulated as

$$\mathcal{L}_m(\boldsymbol{w}_m^t) = \sum_{j=1}^{J} \frac{B_j}{2B} \log\det\left(\frac{1}{B_j}\boldsymbol{Z}_{m,j}\boldsymbol{Z}_{m,j}^* + \frac{\epsilon^2}{d}\boldsymbol{I}\right) - \frac{1}{2}\log\det\left(\frac{1}{B}\boldsymbol{Z}_m\boldsymbol{Z}_m^* + \frac{\epsilon^2}{d}\boldsymbol{I}\right). \tag{7}$$

which essentially drives the feature subspace of each class, represented in the covariance, to be distinct and even orthogonal, thereby facilitating discrimination. This aligns with the theory of coding rate reduction in classification Yu et al. (2020).

*Remark* 3.1. The robustness against noisy labels in our lossy learning objective can be attributable to two dimensions of relaxation: (1) The model is driven to learn the statistical structure of the feature space. We no longer overemphasize the exact values of features, which correspond to mean statistics. Instead, we allow the features to exhibit structured divergence, as in Gaussian-like distributions. It is sufficient for discrimination to capture useful regularities from this divergence, namely by exploring the covariance statistics. (2) The lossy representation by perturbing the feature covariance is de facto spherizing the ellipsoid feature subspace of each class, which may relax the class decision boundaries. That is, while we still endeavor to maintain the maximal information of the given label information, we obtain a resilient feature output that may slightly lean toward another class based on feature subspace interweaving, which is compressed from the original data. See Appendix B for a detailed interpretation with toy examples of the information-theoretic lossy learning objective. Corresponding numerical analyses are provided in Appendix D.1.

## 4 FEDERATED CLASSIFIER VIA COVARIANCE AGGREGATION

**Intrinsic MAP Classifier.** With the features encoded, we proceed to construct a classifier for prediction and subsequent correction. After training under the objective function in Eq. (7), the encoded features are naturally structured as GM clusters. This obviates the need for a separate neural network classifier. Instead, we leverage the probabilistic structure of the features, as defined by Eq. (2) and further reinforced through feature learning, to design a white-box classifier. Given the parameter $\boldsymbol{\theta}^t = \{\pi_j^t, \mathbf{\Sigma}_j^t\}_{j=1}^J$, classification can be performed directly by evaluating the likelihood of each Gaussian component (i.e., each class) and selecting the class with the highest posterior probability. Thus, $\boldsymbol{\theta}^t$ constitutes exactly the intrinsic classifier under Gaussian Discriminant Analysis (GDA) Hastie & Tibshirani (1996). Nevertheless, the distributed datasets remain on local devices for privacy

and communication efficiency, thus all encoded features $\boldsymbol{z}_m$ are supposed to remain local as well. Consequently, our system operates as a federated classifier: each client independently estimates its own local classifier parameters $\boldsymbol{\theta}_m^t$ based on its locally available features, which are subsequently aggregated to construct a global classifier $\boldsymbol{\theta}^t$. In the $t$-th communication round, each device $m$ estimates the local classifier $\boldsymbol{\theta}_m^t = \{\pi_{m,j}^t, \boldsymbol{\Sigma}_{m,j}^t\}_{j=1}^J$ by maximum likelihood estimation (MLE) as

$$\pi_{m,j}^t = \frac{D_{m,j}}{D_m}, \quad \boldsymbol{\Sigma}_{m,j}^t = \frac{1}{D_{m,j}} \sum_{i=1}^{D_{m,j}} \boldsymbol{z}_m^i \boldsymbol{z}_m^{i*} + \frac{\epsilon^2}{d} \boldsymbol{I}, \tag{8}$$

for class $j$, where $\boldsymbol{z}_m^i$ is the feature vector of the $i$-th sample on device $m$ in communication round $t$, $D_{m,j}$ is the size of data of $j$-th class in device $m$, and $D_m$ is the local dataset size in device $m$. After local estimation, each device uploads the local classifier $\boldsymbol{\theta}_m^t$ to the server. The server then aggregates these local classifiers to obtain the global classifier $\boldsymbol{\theta}^t = \{\pi_j^t, \boldsymbol{\Sigma}_j^t\}_{j=1}^J$ as

$$\pi_j^t = \sum_{m=1}^M \frac{D_m}{\sum_{m=1}^M D_m} \pi_{m,j}^t, \quad \boldsymbol{\Sigma}_j^t = \sum_{m=1}^M \frac{D_{m,j}}{\sum_{m=1}^M D_{m,j}} \boldsymbol{\Sigma}_{m,j}^t, \tag{9}$$

for class $j$, as the Gaussian message combining. Note that the privacy issue in communicating $\boldsymbol{\theta}_m^t$ can be ensured for two reasons: 1) It merely characterizes the drastically dimension-reduced features instead of the raw data distributions. 2) The addition of the error tolerance term to the covariance matrix in Eq. (8) further obscures the information. Upon receiving the global classifier, the classification based on the maximum a *posteriori* (MAP) principle can be conducted as

$$\hat{y} = \arg\max_j \left\{ p(y = j \mid \boldsymbol{z}) \propto \pi_j^t \mathcal{N}(\boldsymbol{0}, \boldsymbol{\Sigma}_j^t) \right\}. \tag{10}$$

**Subspace-Augmented Classifier.** The design of the MAP classifier seamlessly aligns with the training objective, where both endeavor to empower discrimination in the Gaussian-like feature subspaces. By expanding the Gaussian distribution and taking the logarithm, Eq. (10) indicates the probability of $\boldsymbol{z}$ belonging to the $j$-th class is proportional to $\log \pi_j^t - \frac{1}{2} \log \|\boldsymbol{\Sigma}_j^t\| - \frac{1}{2} \boldsymbol{z}^\top (\boldsymbol{\Sigma}_j^t)^{-1} \boldsymbol{z}$. As the objective function in Eq. (7) enables discrimination by the subspace orthogonality of different classes, the volume of different feature subspaces can be assumed equal to compare fairly, such that $\|\boldsymbol{\Sigma}_u\| = \|\boldsymbol{\Sigma}_v\| \forall u, v \in [J]$. Meanwhile, the noisy labels introduce direct estimation bias in $\pi_j^t$, which is not regularized during training. Therefore, we resort to the Mahalanobis distance term $\boldsymbol{z}^\top (\boldsymbol{\Sigma}_j^t)^{-1} \boldsymbol{z}$ and generalize it to be a subspace-augmented version as

$$p(y = j \mid \boldsymbol{z}) \propto - \left( \boldsymbol{z}^\top (\boldsymbol{\Sigma}_j^t)^{-\alpha} \boldsymbol{z} \right)^{\frac{1}{\alpha}}, \tag{11}$$

where $\alpha$ is the augmentation coefficient. A larger $\alpha$ generally leads to greater discriminative power in general. Against label noise, setting appropriate $\alpha$ involves a tradeoff between stronger discrimination and higher tolerance to noisy labels. Further discussions on it can be found in Appendix D.2.

**External Corrector.** Once aggregated, the global classifier $\boldsymbol{\theta}^t = \{\boldsymbol{\Sigma}_j^t\}_{j=1}^J$ can be broadcast to the edge devices for label correction over local datasets. We intend to utilize $\boldsymbol{\theta}^t$ to retrospectively classify the training samples to detect noisy labels and correct those with high noise possibility in the correction rounds. Based on the principle of cross-validation Ovi et al. (2023); Chen et al. (2019); Berrar et al. (2019), for each local device, we extract the external information from the global classifier to serve as the external corrector. Specifically, device $m'$'s external corrector $\boldsymbol{\theta}_{\backslash m'}^t = \{\boldsymbol{\Sigma}_{j \backslash m'}^t\}_{j=1}^J$ is calculated as

$$\boldsymbol{\Sigma}_{j \backslash m'}^t = \frac{\sum_{m=1}^M D_{m,j} \boldsymbol{\Sigma}_j^t - D_{m',j} \boldsymbol{\Sigma}_{m',j}^t}{\sum_{m=1}^M D_{m,j} - D_{m',j}}. \tag{12}$$

We can validate the local dataset $\mathcal{D}_m$ based on its corresponding external corrector $\boldsymbol{\theta}_{\backslash m}^t$ and predict as in Eq. (11) to receive $\hat{y}$ and its corresponding probability $p(y = \hat{y} \mid \boldsymbol{z}) \propto - \left( \boldsymbol{z}^\top (\boldsymbol{\Sigma}_{\hat{y} \backslash m'}^t)^{-\alpha} \boldsymbol{z} \right)^{\frac{1}{\alpha}}$. Sample relabeling is performed as

$$(\boldsymbol{x}, y) \leftarrow \{ (\boldsymbol{x}, \hat{y}) \mid p(y = \hat{y} \mid \boldsymbol{z}) \geq \eta_c \,\&\, \hat{y} \neq y \}, \tag{13}$$

where $\eta_c$ is a threshold of confidence. Note that from feature encoding, classifying, and correcting, we consistently follow the key point of building and utilizing a robust discriminative feature space. The covariance matrix is core to representing the feature space, which is exchanged between edge devices and the server for a federated alignment. The whole algorithm is summarized as in Algorithm 1.

---

**Algorithm 1** FedCova

---

1: **Input**: Number of devices $M$, number of classes $J$, training round $T$, correction rounds $\{T_c\}$, $\{\mathcal{D}_m^t\}_{m=1}^M$, dataset size $D = \sum_m D_m$, and learning rate $\eta$.
2: **Initialize**: $t = 0$ and the global model $\boldsymbol{w}^{(0)}$.
3: **for** $t \in [0, T]$ **do**
4:     The server broadcasts the global model $\boldsymbol{w}^t$ and the global classifier $\boldsymbol{\theta}^t = \{\boldsymbol{\Sigma}_j^t\}_{j=1}^J$ to devices;
5:     **for** $m \in [M]$ **do**
6:         Receiving $\boldsymbol{w}^t$ and $\boldsymbol{\theta}^t = \{\boldsymbol{\Sigma}_j^t\}_{j=1}^J$ from the server.
7:         **if** $t \in \{T_c\}$ **then**
8:             Device $m$ computes its external corrector $\boldsymbol{\theta}_{\backslash m}^t = \{\boldsymbol{\Sigma}_{j\backslash m}^t\}_{j=1}^J$ based on Eq. (12);
9:             **if** $p(y = \hat{y} \mid \boldsymbol{z}) \geq \eta_c$ & $\hat{y} \neq y$ **then**
10:                Relabel the sample $(\boldsymbol{x}, y) \leftarrow (\boldsymbol{x}, \hat{y})$ ;
11:             **end if**
12:         **end if**
13:         Device $m$ update local model based on $\boldsymbol{w}_m^{t+1} \leftarrow \boldsymbol{w}^t - \eta \nabla \mathcal{L}(\boldsymbol{w}^t; \mathcal{D}_m^t)$;
14:         Device $m$ outputs local features as $\boldsymbol{z}_m = f_{\boldsymbol{w}^{t+1}}(\boldsymbol{x}_m)$ over $\mathcal{D}_m$;
15:         Device $m$ estimates its local classifier $\boldsymbol{\theta}_m^{t+1} = \{\boldsymbol{\Sigma}_{m,j}^{t+1}\}_{j=1}^J$ based on local features;
16:         Device $m$ uploads its local model $\boldsymbol{w}_m^{t+1}$ and its local classifier $\boldsymbol{\theta}_m^{t+1}$ to the server;
17:     **end for**
18:     The server aggregates the global model $\boldsymbol{w}^{(t+1)} = \sum_m \frac{D_m}{D} \boldsymbol{w}_m^{t+1}$;
19:     The server aggregates the global classifier $\boldsymbol{\theta}^{t+1} = \{\boldsymbol{\Sigma}_j^{t+1}\}_{j=1}^J$ based on Eq. (9);
20: **end for**

---

# 5 SIMULATION RESULTS

## 5.1 EXPERIMENTAL SETUP

**Federated Learning System.** We evaluate the performance of our algorithm on the image classification task over three datasets, CIFAR-10, CIFAR-100 Krizhevsky et al. (2009), and a real-world noisy dataset, Clothing1M Xiao et al. (2015). For the network architecture, we employ Resnet-18 He et al. (2016) for CIFAR-10, Resnet-34 for CIFAR-100, and Resnet-50 for Clothing1M. Complete setup details are provided in Appendix C.1.

**Noise Setting and Data Heterogeneity.** For datasets other than Clothing1M, we introduce a bi-level noise scheme to simulate realistic noise scenarios in both symmetric and asymmetric noise patterns. This setup is characterized by the noisy device ratio, denoted as $\rho$, and the sample noise ratio, denoted as $\tau$. In the main experiments, we set the noise-pairs of $(\rho, \tau)$ from $\{(0.4, 0.5), (0.4, 0.7), (0.6, 0.5), (0.6, 0.7), (0.8, 0.5), (0.8, 0.7)\}$ in symmetric noise pattern and $\{(0.4, 0.3), (0.4, 0.5), (0.4, 0.7), (0.6, 0.3), (0.6, 0.5), (0.6, 0.7)\}$ in asymmetric noise pattern. For the data heterogeneity setting, we use a general distribution set that considers both the class heterogeneity and the dataset size heterogeneity, which are simulated by a Bernoulli distribution with a probability $p$ and a Dirichlet distribution parameterized by $\alpha_{\text{Dir}} > 0$, respectively. We set $p = 0.5$ and $\alpha_{\text{dir}} = 5$ as a relatively high non-i.i.d. scenario.

**Baselines.** We evaluate the proposed algorithms compared with several state-of-the-art (SOTA) frameworks for FL under noisy labels, including FedNed Lu et al. (2024), FedNed- Lu et al. (2024), FedNoRo Wu et al. (2023b), RoFL Yang et al. (2022), FedCorr Xu et al. (2022), and FedAvg McMahan et al. (2017). Meanwhile, typical algorithms for noisy label learning from centralized learning, Co-teaching Han et al. (2018) and DivideMix Li et al. (2020), have also been compared under our federated settings. Complete setup details for baselines are provided in Appendix C.1.

## 5.2 PERFORMANCE COMPARISON

Table 1 provides detailed analyses of model accuracy on non-i.i.d. CIFAR-10 dataset under different settings of symmetric noisy levels. On the whole, FedCova achieves the highest test accuracy across all noise levels. When the noise is relatively low, e.g., $(\rho, \tau) = (0.4, 0.5)$, most baselines remain robust,

Table 1: The average accuracy and standard deviation of the last five rounds on CIFAR-10 (non-i.i.d.) at different noise levels ($\rho$: noisy device ratio, $\tau$: sample noise ratio) of symmetric noise.

| Method | Test Accuracy (%) ± Standard Deviation (%) | | | | | |
| --- | --- | --- | --- | --- | --- | --- |
| | $\rho = 0.4$ | | $\rho = 0.6$ | | $\rho = 0.8$ | |
| | $\tau = 0.5$ | $\tau = 0.7$ | $\tau = 0.5$ | $\tau = 0.7$ | $\tau = 0.5$ | $\tau = 0.7$ |
| FedAvg | 72.43±4.63 | 65.16±8.36 | 64.75±1.46 | 52.87±3.68 | 47.98±5.42 | 22.27±5.04 |
| CoteachingFL | 78.63±0.58 | 73.58±0.78 | 73.34±0.67 | 65.00±0.72 | 57.41±0.81 | 38.86±0.58 |
| DivideMixFL | 68.50±0.06 | 64.58±0.03 | 66.32±0.16 | 65.42±0.18 | 59.88±0.14 | 53.16±0.05 |
| RoFL | 77.78±0.23 | 72.79±0.21 | 73.52±0.35 | 62.51±0.79 | 58.18±0.14 | 44.11±0.25 |
| FedCorr | 84.87±0.66 | 79.24±0.44 | 69.47±0.32 | 67.21±1.69 | 61.74±1.86 | 48.15±1.38 |
| FedNoRo | 81.51±0.33 | 80.48±0.30 | 74.05±0.15 | 71.93±0.22 | 63.79±1.20 | 30.23±1.02 |
| FedNed- | 72.90±1.40 | 64.30±1.01 | 70.15±1.29 | 65.71±1.45 | 52.33±1.82 | 39.63±1.88 |
| FedNed | 82.38±0.13 | 82.02±0.56 | 78.16±1.23 | 77.11±0.45 | 64.58±1.85 | 48.98±1.39 |
| **FedCova** | **86.52±0.23** | **85.50±0.38** | **83.78±0.68** | **80.71±0.56** | **67.21±0.92** | **64.99±0.75** |

Table 2: The average accuracy and standard deviation of the last five rounds on CIFAR-10 (non-i.i.d.) at different noise levels of asymmetric noise.

| Method | Test Accuracy (%) ± Standard Deviation (%) | | | | | |
| --- | --- | --- | --- | --- | --- | --- |
| | $\rho = 0.4$ | | | $\rho = 0.6$ | | |
| | $\tau = 0.3$ | $\tau = 0.5$ | $\tau = 0.7$ | $\tau = 0.3$ | $\tau = 0.5$ | $\tau = 0.7$ |
| FedAvg | 72.06±4.35 | 71.23±5.86 | 67.40±11.2 | 68.91±5.02 | 63.53±2.89 | 43.40±18.2 |
| RoFL | 78.87±0.10 | 77.85±0.53 | 71.56±0.46 | 78.45±0.32 | 66.10±2.68 | 41.38±1.00 |
| FedCorr | 75.21±0.57 | 62.66±1.18 | 38.67±1.23 | 39.37±0.49 | 34.00±0.29 | 30.66±0.25 |
| FedNoRo | 86.40±0.72 | 85.72±0.41 | 84.76±0.62 | 75.27±0.21 | 63.88±19.9 | 33.42±37.8 |
| FedNed- | 75.69±0.98 | 70.38±1.40 | 65.51±3.48 | 78.63±0.37 | 71.22±1.15 | 66.44±1.44 |
| FedNed | 82.30±0.35 | 79.73±0.43 | 78.54±0.65 | 79.77±0.57 | 78.03±0.34 | 74.67±0.66 |
| **FedCova** | **88.00±0.25** | **87.84±0.24** | **87.77±0.19** | **87.83±0.40** | **87.31±0.39** | **87.29±0.26** |

and methods like FedCorr can still exploit enough clean devices to guide training. However, once clean devices no longer dominate (i.e., $\rho \geq 0.5$), their detection becomes unreliable and performance drops sharply; for instance, FedCorr declines from 79.24% at $(\rho, \tau) = (0.4, 0.7)$ to 69.47% at $(0.6, 0.5)$ despite similar amounts of noisy data. As noise further increases, FedCova maintains strong performance across broader settings, underscoring its system-level robustness. This advantage stems from its covariance-based feature alignment, which yields stable representations, whereas most baselines rely on cross-entropy that still fits noisy labels. FedNed reduces noise impact via negative distillation and performs well when noisy devices are few, maintaining above 85% at $\rho = 0.4$ and around 77–78% at $\rho = 0.6$, but it degrades under higher noisy-device ratios. RoFL benefits from feature alignment, yet its centroid aggregation only captures mean statistics and remains noise-sensitive. In contrast, FedCova leverages the covariance structure, fully manifesting the discriminative power of the feature space. Numerical comparison results in the asymmetric noise pattern are summarized in Table 2. Under such a serious asymmetry, baselines like FedCorr and FedNoRo are more sensitive to high noise levels. In contrast, FedCova demonstrates superior robustness under noisy labels. As the noise level changes from $(0.4, 0.3)$ to $(0.6, 0.7)$, the test accuracy is guaranteed around 87–88%. This can inherently be attributed to the orthogonality structure of the class subspaces driven by FedCova. When in asymmetric noise, the tangle is over fewer classes, extremely between two classes, the subspaces are then only mixed mainly over two classes, which is more comfortable for the discrimination in FedCova.

Experiments on CIFAR-100 with more classes intricately intermingled further intensify the challenge of FL algorithms under noisy labels. Results can be observed as in Table 3. The discrimination boundaries between classes become further complicated to be learned as the number of classes increases. Similarly to the former analysis, FedCova still outperforms baselines under various settings of the noisy levels. Additionally, we evaluate the algorithms under a real-world dataset with

Table 3: The average accuracy and standard deviation on CIFAR-100 (non-i.i.d.) at different noise levels of symmetric noise.

| Method | Test Accuracy (%) $\pm$ Standard Deviation (%) | | | | | |
| | $\rho = 0.4$ | | $\rho = 0.6$ | | $\rho = 0.8$ | |
| | $\tau = 0.5$ | $\tau = 0.7$ | $\tau = 0.5$ | $\tau = 0.7$ | $\tau = 0.5$ | $\tau = 0.7$ |
|---|---|---|---|---|---|---|
| FedAvg | 57.70$\pm$2.95 | 51.21$\pm$1.47 | 41.65$\pm$4.32 | 31.58$\pm$6.38 | 38.49$\pm$2.37 | 18.27$\pm$5.24 |
| RoFL | 50.71$\pm$0.57 | 48.80$\pm$0.16 | 45.58$\pm$0.17 | 37.89$\pm$0.28 | 29.40$\pm$0.62 | 5.20$\pm$0.03 |
| FedCorr | 59.66$\pm$0.93 | 55.60$\pm$0.45 | 53.81$\pm$0.60 | 50.01$\pm$0.98 | 49.68$\pm$0.46 | 29.04$\pm$0.61 |
| FedNoRo | 62.20$\pm$0.47 | 58.57$\pm$0.50 | 42.86$\pm$0.16 | 26.70$\pm$1.08 | 24.43$\pm$0.31 | 14.13$\pm$0.26 |
| FedNed- | 57.56$\pm$0.74 | 55.35$\pm$0.38 | 51.83$\pm$0.86 | 46.37$\pm$0.63 | 43.44$\pm$1.29 | 32.65$\pm$0.60 |
| FedNed | 64.67$\pm$0.40 | 61.53$\pm$2.09 | 56.63$\pm$0.45 | 54.84$\pm$0.60 | 49.51$\pm$0.95 | 36.04$\pm$0.33 |
| **FedCova** | **67.63$\pm$1.04** | **66.84$\pm$0.73** | **60.67$\pm$1.11** | **58.46$\pm$1.58** | **55.71$\pm$0.82** | **40.80$\pm$0.78** |

noisy labels, Clothing1M, as shown in Table 4. FedCova also achieves the highest accuracy compared with baselines.

Table 4: The average accuracy and standard deviation on Clothing1M (non-i.i.d.).

| | FedAvg | RoFL | FedCorr | FedNoRo | FedNed | **FedCova** |
|---|---|---|---|---|---|---|
| Accuracy (%) | 41.07$\pm$1.71 | 59.75$\pm$0.56 | 56.66$\pm$0.23 | 36.42$\pm$0.27 | 55.80$\pm$0.11 | **61.42$\pm$0.65** |

Table 5 lists the additional conditions of the baselines, which describe the extra dependencies that incur a non-negligible resource consumption. FedCova achieves the competitive performance without introducing additional resource dependencies. The baselines FedCorr and FedNoRo rely on local warm-up training, which entails hundreds of device–server exchanges and consequently much higher communication overhead than standard federated training. Nevertheless, their ablation studies identify warm-up as the most effective component. FedNed- and FedNed rely on a clean public dataset as a benchmark for training. In addition, FedNed necessitates the presence of extremely noisy devices to identify what should not be learned, which limits its applicability to certain noisy environments. Instead, our framework internally explores robustness within the model itself and leverages the structure of the feature space, making it applicable to a wide range of noisy scenarios without the need for ideal benchmarks. Further experimental results on subspace orthogonality and correction performance are referred to Appendix C.

Table 5: Additional conditions. WU: warm-up communication rounds; CD: additional clean public dataset; ED: extremely noisy device.

| | FedAvg | RoFL | FedCorr | FedNoRo | FedNed- | FedNed | **FedCova** |
|---|---|---|---|---|---|---|---|
| Additional Conditions | - | - | WU | WU | CD | CD, ED | - |

## 5.3 ABLATION STUDY

We evaluate the performance of FedCova with and without several key designs, considering the following configurations: "FedCova w/o ext. corrector", FedCova without label correction by the external corrector; "FedCova w/o subs. augment.", FedCova with the subspace augmentation coefficient set to $\alpha = 1$, reducing the method to the Mahalanobis distance as used in the GM-based MAP classifier; "FedCova w/o classifier align.", FedCova without federated Gaussian message combining of covariance matrices, instead aggregating them with equal weights; "FedCova w/o error tolerance", FedCova with the error tolerance coefficient set to a small value, rendering the feature subspace relaxation ineffective; "FedCova w/o zero mean", FedCova removing the zero mean assumption and involve mean statistics in aggregation, resulting in a mean-aware classifier.

Table 6 summarizes the ablation study results on non-i.i.d. CIFAR-10 dataset under the noisy level $(\rho, \tau) = (0.6, 0.7)$. All components are validated to contribute to improving accuracy. The focus on feature covariances is the key advantage of FedCova, as demonstrated by the results of "FedCova w/o zero mean" and "FedCova w/o subs. augment." "FedCova w/o zero mean" especially shows that the bias of means introduced by noisy labels leads to degraded performance. "FedCova w/o error tolerance" exhibits the largest performance gap, likely because the strict orthogonality enforced by our mutual information maximization objective may

Table 6: Effect of FedCova's key components on CIFAR-10 (non-i.i.d.) under the noise level $(\rho, \tau) = (0.6, 0.7)$.

| Configuration | Accuracy (%) |
|---|---|
| FedCova | 80.71±0.56 |
| FedCova w/o ext. corrector | 77.88±1.47 |
| FedCova w/o subs. augment. | 79.58±0.60 |
| FedCova w/o classifier align. | 76.86±0.30 |
| FedCova w/o error tolerance | 69.47±0.51 |
| FedCova w/o zero mean | 76.93±0.34 |

vary across local training processes. Introducing the error-tolerance identity matrix both spherizes feature subspaces to resist noisy labels and aligns representation orthogonality across devices. Similarly, the significance of aligning representations across devices is further evidenced in "FedCova w/o classifier align.". "FedCova w/o ext. corrector" highlights the effectiveness of our corrector design as well as the discriminative capability of the covariance-based classifier. Further addition experiments and analysis on the effect of several hyperparameters are provided in Appendix D.

## 6 CONCLUSION

We have proposed FedCova, a covariance-aware federated learning framework tailored to tackle the challenges posed by noisy labels in distributed datasets. By exploiting the robustness of feature representations captured by covariances, FedCova seamlessly integrates feature encoding, intrinsic classifier construction, and noisy label correction into a unified framework. The mutual information maximization objective constrains error-tolerant covariance structures, enabling the model to learn discriminative and orthogonal feature subspaces that enhance resilience to label noise. Through covariance aggregation, FedCova effectively aligns federated classifiers across edge devices. By introducing subspace augmentation, a generalized classifier is derived based on intrinsic MAP discrimination. Additionally, local external correctors facilitate effective relabeling while mitigating self-bias. Extensive experiments over several datasets demonstrated that FedCova consistently outperforms state-of-the-art frameworks in scenarios under various levels of label noise, highlighting its robustness.

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

# A RELATED WORKS

## A.1 FEDERATED LEARNING UNDER NOISY LABELS

As the scale of training data increases rapidly, the issue of noisy labels—stemming from annotation errors, sensor faults, or adversarial attacks—has become increasingly prominent. Attention is first paid to centralized learning with noisy labels Northcutt et al. (2021); Chen et al. (2019). Mainstream solutions can be broadly categorized into two types. One category refines the training process by incorporating offline techniques such as noise detection, sample selection, and label correction, aiming to design noise-resistant algorithms for a given training network Cheng et al. (2022); Ren et al. (2018); Englesson & Azizpour (2021); Xia et al. (2024). JointOpt Tanaka et al. (2018) is a classic example that introduces a joint optimization framework to correct the labels of noisy samples during training. This is accomplished by alternating between updating the model parameters and refining the labels. The other category expands the learning structure by introducing additional resource conditions, such as assuming access to clean public datasets, duplicating the network for cross-validation, or utilizing advanced frameworks like unsupervised learning and meta-learning, etc. Chen et al. (2019); Li et al. (2022); Wu et al. (2021); Zheng et al. (2024). Co-teaching Han et al. (2018) is a representative framework that simultaneously trains two networks, where each network selects small batches of clean samples to update the other. By relying on peer networks to filter our noisy labels, Co-teaching effectively reduces the impact of label noise and improves robustness. DivideMix Li et al. (2020) further advances this approach by dynamically dividing the training dataset with label noise into clean and noisy subsets while incorporating auxiliary semi-supervised learning algorithms.

When faced with distributed data and privacy constraints in federated learning (FL), the server cannot directly access local data samples, which limits the applicability and effectiveness of these robust techniques developed for centralized learning in FL. Consequently, researchers have focused on developing approaches tailored to the federated paradigm. Leveraging the native partitioning of devices in federated learning enables methods for sample selection and noise filtering Wu et al. (2023a). For instance, FedCorr Xu et al. (2022) proposes a dimensionality-based noise filtering strategy, dividing clients into clean and noisy groups using local intrinsic dimensionality measures, and further trains local filters to distinguish clean samples from noisy clients via per-sample training loss analysis. Similarly, FedDiv Li et al. (2024a) enhances noise filtering by aggregating shared filter knowledge from clients, thereby improving the detection of label noise within each client's dataset. However, these approaches remain constrained by their reliance on the output-label relationship, as their learning objectives–such as cross-entropy loss–continue to bind the model to potentially incorrect labels. Recent studies have sought to overcome these limitations by introducing novel schemes for noisy label detection or correction. FedFixer Ji et al. (2024) equips each device with two replicated local models that perform cross-learning, enabling collaborative mitigation of label noise and reduction of overfitting. FedNed Lu et al. (2024) applies negative distillation from extremely noisy devices to motivate the model away from incorrect directions with a new loss function designed based on Kullback-Leibler (KL) divergence. Similar redesign of loss functions to mitigate the aforementioned overfitting in objectives is also proposed as FedNoRo Wu et al. (2023b) and label-noise-aware distillation Tsouvalas et al. (2024). While effective, these methods require additional conditions such as additional duplicate models or public clean datasets and incur increased computational overhead, which may limit their scalability in complex label noise scenarios. In contrast, our framework first addresses the conventional output-label binding by redesigning the loss function and further eliminates the reliance on supplementary conditions, inherently guiding the learning model to achieve system-level robustness against noisy labels.

## A.2 FEDERATED FEATURE LEARNING

Learning and understanding features is crucial for uncovering underlying data structures and improving model generalization. Conventional representation learning emphasizes the importance of acquiring discriminative and informative feature representations that capture the intrinsic structure of data Bengio et al. (2013); Han et al. (2020). By transforming raw data into a feature space where the underlying patterns and relationships are more apparent, learning such features enables models to focus on the statistical properties of the data to achieve a competitive performance Yu et al. (2020); Li et al. (2024b); Yu et al. (2021). This opens up opportunities to enhance robustness against

noisy labels, since promoting feature learning that enables a genuine understanding of the input data makes the model less directly influenced by incorrect labels Arpit et al. (2017); Zhang et al. (2023). Building on this insight, approaches like contrastive learning Li et al. (2021); Yi et al. (2022) have shown considerable success in reducing the adverse effects of noisy labels within centralized learning frameworks. By penalizing feature similarity in pairwise loss between samples, contrastive learning encourages the model to capture class-discriminative representations, thereby enhancing robustness to label noise.

However, directly applying these methods to federated learning (FL) is challenging, since data remains isolated on individual devices and direct access to sample pairs across devices is not allowed. Consequently, approaches like contrastive learning, which require contrasting all sample pairs, often underperform in FL scenarios Zhang et al. (2023); Duan et al. (2022). In FL, learning informative features has been primarily explored through the lens of federated representation learning and prototype-based methods. For instance, pFedFDA Mclaughlin & Su (2024) proposes a novel approach to personalized FL by framing representation learning as a generative modeling task, enabling efficient adaptation of global classifiers to local feature distributions. FURL Bui et al. (2019) introduces a privacy-preserving and resource-efficient approach in FL by locally training private user embeddings. To achieve global consistency among local representations, some works, such as FedProto Tan et al. (2022) and FedPAC Xu et al., aggregate the local means (centroids) of the learned features and introduce a regularization term in the loss function for alignment. While these methods underscore the potential of feature learning in FL, they primarily address data heterogeneity and often do not explicitly tackle the challenge of noisy labels. Moreover, they remain vulnerable to overfitting caused by the output-label dependency inherent in the cross-entropy loss. Additionally, the means of features estimated under noisy labels can be significantly biased, resulting in inaccurate noise detection and further aggravating overfitting.

To address noisy labels through feature utilization, FedCoop Tam et al. (2023) proposes a cooperative FL framework that leverages cross-client collaboration and feature-based quality assessment to perform semi-supervised training, thereby robustly mitigating the impact of noisy labels. However, it primarily focuses on filtering noisy samples by relying on clean subsets, rather than leveraging the intrinsic structure of the feature space. Fed-DR-Filter Duan et al. (2022) introduces a novel filtering method in FL, which identifies clean data based on global data representation correlations. However, directly transmitting all features from clients to the server leads to excessive communication overhead and raises privacy concerns, while the server also bears significant computational costs for noise filtering. RoFL Yang et al. (2022) mitigates label noise in FL by aligning class-wise feature centroids (means) between the server and clients, thus promoting consistent decision boundaries across local models. It further combines sample selection and label correction to enhance robustness against noisy annotations. Nevertheless, as previously noted, the reliance on mean statistics for alignment leaves the method sensitive to label noise, particularly under cross-entropy loss, as mean features can be heavily biased by mislabeled data. Our framework instead emphasizes the covariance statistics and designs a unified system that spans from feature encoding to discriminative classification and label correction, relieving the aforementioned restrictions while enhancing robustness.

## B UNDERSTANDING THE LOSSY LEARNING OBJECTIVE

In this section, we provide a further statement for the lossy learning objective design in Section 3.

### B.1 MUTUAL INFORMATION MAXIMIZATION

The core idea behind our objective function in Eq. (7) is to maximize the mutual information between the learned representation $Z$ and its associated label $Y$. Here we provide the following step-by-step explanation.

Mutual information quantifies how much knowing one random variable reduces the uncertainty about another. Formally, for random variables $Z$ and $Y$, the mutual information is defined as:

$$I(Z;Y) = h(Z) - h(Z|Y), \tag{14}$$

where $h(Z)$ is the differential entropy of $Z$, and $h(Z|Y)$ is the conditional entropy of $Z$ given $Y$. In words, $h(Z)$ measures the total uncertainty (randomness) in $Z$, $h(Z|Y)$ measures the uncertainty in

$Z$ after knowing $Y$, and their difference, $I(Z;Y)$, captures the reduction in uncertainty about $Z$ due to knowledge of $Y$.

The approach of leveraging such mutual information to drive the model training is formalized as the *Information Bottleneck* (IB) principle Tishby et al. (2000); Tishby & Zaslavsky (2015). The IB framework proposes learning a representation $Z$ of the input $X$ that compresses $X$ as much as possible while still preserving information about the target $Y$. This trade-off is typically expressed as maximizing $I(Z;Y) - \beta I(Z;X)$, where $\beta \geq 0$ balances sufficiency (informativeness for $Y$) and minimality (compression of $X$). As we aim to learn class-discriminative features within a federated feature learning framework, our objective in Eq. (3) is to maximize the mutual information between the learned representation $Z$ and the label $Y$, ensuring that the features capture as much relevant information about the target as possible. In other words, we want $Z$ to be highly predictive of $Y$. By maximizing $I(Z;Y)$, we ensure that the representation $Z$ retains the aspects of the input $X$ that are most informative for predicting $Y$, while potentially discarding irrelevant variations. In our case, we focus solely on maximizing $I(Z;Y)$, which can be seen as a special case of the IB principle by setting $\beta = 0$; that is, we prioritize learning representations that are maximally informative about $Y$. To clarify, we seek to retain the most label-relevant information from $X$ in the encoded feature $Z$, ensuring that $Z$ is sufficiently informative for accurate classification.

Suppose $Y$ is a discrete class label taking values in $\{1, \ldots, J\}$ with probability $\pi_j = P(Y = j)$. Based on the Gaussian mixture prior defined in Eq. (2), for each class $j$, the conditional distribution of the representation is given by

$$Z \mid Y = j \sim \mathcal{N}(\mathbf{0}, \mathbf{\Sigma}_j).$$

The conditional entropy $h(Z|Y)$ quantifies the expected uncertainty in $Z$ given knowledge of $Y$. Since $Z|Y = j$ follows a multivariate Gaussian distribution for each $j$, the differential entropy for a $d$-dimensional Gaussian $\mathcal{N}(\boldsymbol{\mu}, \mathbf{\Sigma}_j)$ is

$$h(Z|Y = j) = \frac{1}{2} \log \left[ (2\pi e)^d \det(\mathbf{\Sigma}_j) \right]. \tag{15}$$

Notably, the entropy depends only on the covariance matrix $\mathbf{\Sigma}_j$, not on the mean $\boldsymbol{\mu}$. Therefore, the conditional entropy of $Z$ given $Y$ is the expectation over all classes as

$$h(Z|Y) = \sum_{j=1}^{J} \pi_j \, h(Z|Y = j) = \frac{1}{2} \sum_{j=1}^{J} \pi_j \log \left[ (2\pi e)^d \det(\mathbf{\Sigma}_j) \right]$$

$$= \frac{1}{2} \sum_{j=1}^{J} \pi_j \log \det(\mathbf{\Sigma}_j) + \frac{d}{2} \log(2\pi e). \tag{16}$$

The marginal distribution of $Z$ thus becomes a mixture of Gaussians. For analytic tractability, we often approximate $p(\boldsymbol{z})$ as a single Gaussian with zero mean and covariance $\mathbf{\Sigma}$. Thus, the marginal entropy $h(Z)$ can be computed as

$$h(Z) = \frac{1}{2} \log \left[ (2\pi e)^d \det(\mathbf{\Sigma}) \right] = \frac{1}{2} \log \det(\mathbf{\Sigma}) + \frac{d}{2} \log(2\pi e). \tag{17}$$

Plugging the entropy formula Eqs. (16) and (17) into the mutual information Eq. (14), we obatin

$$I(Z;Y) = h(Z) - h(Z|Y) = \frac{1}{2} \log \det(\mathbf{\Sigma}) - \frac{1}{2} \sum_{j=1}^{J} \pi_j \log \det(\mathbf{\Sigma}_j). \tag{18}$$

In practice, we minimize its negative as in the objective function in Eq. (4). Since the entropy of a Gaussian variable depends only on its covariance, not its mean, our objective is independent of the feature means and we can focus solely on the covariance matrices.

### B.2 WHY $I(Z;Y)$ WORKS: RELATION TO CROSS-ENTROPY LOSS

Alternatively, we can also rewrite the mutual information between the feature $Z$ and the label $Y$ as

$$I(Z;Y) = H(Y) - H(Y|Z) = H(Y) + \mathbb{E}_{p(X,Y)} \mathbb{E}_{p(Z|X)} \log p(Y|Z). \tag{19}$$

As we maximize the mutual information $I(Z; Y)$, given the constant $H(Y)$, we actually are training under the loss function

$$\mathcal{L}_{MI} = -\mathbb{E}_{p(X,Y)}\mathbb{E}_{p(Z|X)} \log p(Y|Z). \tag{20}$$

When training under conventional objectives like cross-entropy loss, the true conditional distribution $p(Y|Z)$ is typically unknown and is thus replaced by a predictive distribution $\hat{p}(Y|Z)$ parameterized by the model (e.g., a neural network classifier). From the viewpoint of cross-entropy, the general objective is the expected negative log-likelihood under the model, given by

$$\mathcal{L}_{\text{CE}} = H(p(Y|X), \hat{p}(Y|X)) = -\mathbb{E}_{p(X,Y)} \log \hat{p}(Y|X), \tag{21}$$

where $H(p(Y|X), \hat{p}(Y|X))$ denotes the cross-entropy between the true conditional distribution $p(Y|X)$ and the model prediction $\hat{p}(Y|X)$. By the standard decomposition of cross-entropy in terms of entropy and KL divergence, we have

$$H(p(Y|X), \hat{p}(Y|X)) = H(p(Y|X)) + D_{\text{KL}}(p(Y|X)\|\hat{p}(Y|X)) \tag{22}$$

$$= -\mathbb{E}_{p(X,Y)} \log p(Y|X) + \mathbb{E}_{p(X,Y)} \log \frac{p(Y|X)}{\hat{p}(Y|X)}, \tag{23}$$

where $D_{\text{KL}}(p(Y|X)\|\hat{p}(Y|X))$ is the KL divergence between the true and the predicted conditional distributions, which is always non-negative. Therefore, this decomposition implies

$$-\mathbb{E}_{p(X,Y)} \log \hat{p}(Y|X) \geq -\mathbb{E}_{p(X,Y)} \log p(Y|X). \tag{24}$$

Consider the scenario where the predictive distribution is conditioned on a latent variable $Z$ as an intermediate feature representation extracted from $X$, and thus the above results naturally generalize to

$$-\mathbb{E}_{p(X,Y)} \log \mathbb{E}_{p(Z|X)}\hat{p}(Y|Z) \geq -\mathbb{E}_{p(X,Y)} \log \mathbb{E}_{p(Z|X)}p(Y|Z). \tag{25}$$

This inequality shows that the cross-entropy loss $\mathcal{L}_{\text{CE}}$ under the model's predictive distribution is always at least as large as the entropy of the true conditional distribution $p(Y|Z)$.

Furthermore, due to the concavity of the logarithm function, Jensen's inequality gives, for any fixed $(X, Y)$,

$$\mathbb{E}_{p(Z|X)} \log p(Y|Z) \leq \log \mathbb{E}_{p(Z|X)}p(Y|Z). \tag{26}$$

Taking the negative and averaging over the data distribution, we obtain

$$-\mathbb{E}_{p(X,Y)}\mathbb{E}_{p(Z|X)} \log p(Y|Z) \geq -\mathbb{E}_{p(X,Y)} \log \mathbb{E}_{p(Z|X)}p(Y|Z). \tag{27}$$

Therefore, the mutual information-based objective $\mathcal{L}_{\text{MI}}$ is also an upper bound of the entropy of the true conditional distribution $p(Y|Z)$, just as the cross-entropy loss is. Minimizing either $\mathcal{L}_{\text{CE}}$ or $\mathcal{L}_{\text{MI}}$ drives the model to approach the true entropy from different perspectives. While cross-entropy loss focuses on directly improving the predictive distribution of $Y$ given $Z$, the mutual information objective emphasizes the informativeness of the learned feature representation $Z$ about $Y$. As a result, using $\mathcal{L}_{\text{MI}}$ as an objective is reasonable, since it encourages the model to extract features that are maximally informative about the target, which can be beneficial for downstream prediction tasks.

### B.3 INTERPRETATION OF LOSSY REPRESENTATION

The objective function presented in Eq. (4) is carefully designed to regularize the structure of the feature space, thereby enhancing class separability through the principal directions of each class. We normalize the features as $\|\boldsymbol{z}\|_2 = 1$ to focus on the directions between different representations, so that all feature points lie on the surface of a unit sphere in the feature space. The term $\log \det(\boldsymbol{\Sigma})$ reflects the volume of the ellipsoid spanned by the feature distribution, which is proportional to the product of its principal variances. By minimizing the weighted sum of log-determinants of class-specific covariance matrices, i.e., $\sum_{j=1}^{J} \frac{B_j}{2B} \log \det\left(\frac{1}{B_j}\boldsymbol{Z}_{m,j}\boldsymbol{Z}_{m,j}^* + \frac{\epsilon^2}{d}\boldsymbol{I}\right)$, the objective encourages the feature distribution within each class to be as compact as possible, thus reducing intra-class variance. Conversely, maximizing the log-determinant of the overall covariance matrix, i.e., $\frac{1}{2} \log \det\left(\frac{1}{B}\boldsymbol{Z}_m\boldsymbol{Z}_m^* + \frac{\epsilon^2}{d}\boldsymbol{I}\right)$, promotes the expansion of the global feature distribution along as many orthogonal directions as possible, thereby increasing inter-class variance. The objective thus implicitly drives the principal directions of different classes to become mutually orthogonal, as

this configuration maximizes the total volume covered by the union of all class distributions while keeping each class individually tight.

This regularization has a clear information-theoretic interpretation: it balances the reduction of uncertainty within each class against the increase of total uncertainty when all classes are considered together. As a result, the learned feature space is structured such that samples from different classes are distributed along distinct, nearly orthogonal directions, significantly enhancing the discriminability between classes. This structure enables efficient intrinsic classification based on the features, as each class occupies a unique, maximally separated subspace in the feature space. Therefore, the loss in Eq. (4) is theoretically grounded to encourage both intra-class compactness and inter-class orthogonality, leading to improved classification performance and feature interpretability.

Although the objective in Eq. (4) regularizes the structure of features within each class to be distinct along the principal directions of the feature space, it still suffers from a strong dependency between the output feature $z$ and the label $y$ (which may be a potentially noisy label $y'$). When an encoded feature $z$ is assigned an incorrect label $y'$, it is inevitably incorporated into the calculation of $\Sigma_{y'}$. To mitigate this dependency, we relax the discriminative orthogonality of the feature subspaces by introducing an error tolerance term into the estimated covariance as $\hat{\Sigma}$, resulting in a variant loss function as shown in Eq. (7). Mathematically, for the batch covariance $\frac{1}{B}Z_m Z_m^*$, we can express its spectral decomposition as

$$\frac{1}{B}Z_m Z_m^* = U_m \Lambda_m U_m^*, \tag{28}$$

where $U_m$ is the matrix of eigenvectors and $\Lambda_m = \mathrm{diag}(\lambda_{m,1}, \ldots, \lambda_{m,d})$ contains the eigenvalues of the covariance. With the error tolerance term $\frac{\epsilon^2}{d}I$, the estimated covariance can be decomposed as

$$\hat{\Sigma}_m = \frac{1}{B}Z_m Z_m^* + \frac{\epsilon^2}{d}I = U_m \left(\Lambda_m + \frac{\epsilon^2}{d}I\right) U_m^*, \tag{29}$$

which means that each eigenvalue is expanded as

$$\hat{\lambda}_{m,p} = \lambda_{m,p} + \frac{\epsilon^2}{d}. \tag{30}$$

This uniform shift in the eigenvalues effectively smooths the spectrum of the covariance matrix, reducing the dominance of any single principal direction. By introducing this isotropic error tolerance term, the strict orthogonality constraints among class-specific feature subspaces are relaxed, which in turn mitigates the influence of outliers or mislabeled samples and prevents the loss function from being overly sensitive to label noise.

**Toy Example for the Effect of Error Tolerance.**   Here we illustrate a simple toy example to further clarify the effect of the error tolerance term on the feature distributions and the resulting decision boundaries. Specifically, we consider two classes whose feature distributions follow 2-D zero-mean Gaussian distributions, i.e., $\mathcal{N}(\mathbf{0}, \Sigma_A)$ and $\mathcal{N}(\mathbf{0}, \Sigma_B)$, with covariance matrices

$$\Sigma_A = \begin{bmatrix} 4.0 & 0 \\ 0 & 0.2 \end{bmatrix}, \quad \Sigma_B = \begin{bmatrix} 0.2 & 0 \\ 0 & 3.0 \end{bmatrix}.$$

These settings represent two classes that are each elongated along orthogonal directions in the feature space. Fig. 2 visualizes the probability density surfaces for the two classes under three different values of the error tolerance parameter $\epsilon$, where the regularized covariances are given by $\Sigma_A^{(\epsilon)} = \Sigma_A + \frac{\epsilon^2}{d}I$ and $\Sigma_B^{(\epsilon)} = \Sigma_B + \frac{\epsilon^2}{d}I$ with $d = 2$.

Based on the illustration in Fig. 2, we can observe:

- **When $\epsilon^2$ (no error tolerance term):** The two distributions are highly anisotropic and elongated along the $x$- and $y$-axes, respectively. The decision boundary (black contour) is sharply defined at the intersection of the two distributions, and is highly sensitive to the principal directions of the original covariances.
- **As $\epsilon^2$ increases:** The isotropic term begins to smooth out the differences between the two distributions. The shapes become more rounded, the peaks lower, and the decision boundary becomes noticeably smoother and less tightly fitted to the intersection of the original distributions, thus reducing sensitivity to outliers with label noise.

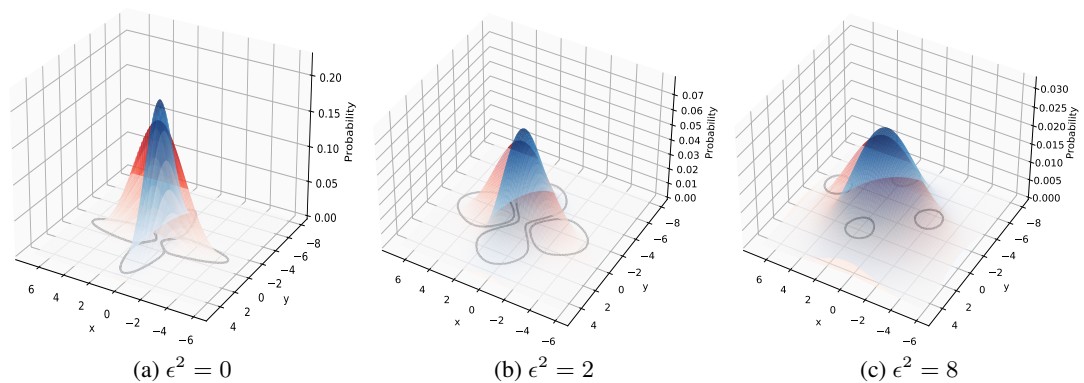

Figure 2: Effect of adding error tolerance term to the covariance matrix on the probability distributions.

- **For large** $\epsilon^2$**:** Both distributions approach nearly isotropic (spherical) shapes, and their density surfaces overlap more substantially. The decision boundary is further smoothed and becomes almost circular, indicating that the strict alignment with the original principal directions is highly relaxed, but tolerance to noise is greatly enhanced.

To further illustrate these effects, Fig. 4 visualizes both the sample distributions in the original feature space and their projections onto the principal axes of each class under varying $\epsilon^2$. As $\epsilon^2$ increases, the samples from both classes become more isotropic, and the distinction between their principal directions diminishes, consistent with the trends observed in the 3-D probability surfaces. The projection histograms show that, when $\epsilon^2 = 0$, each class exhibits a sharply peaked and well-separated distribution along its own principal axis, reflecting strong discrimination. As $\epsilon^2$ grows, the projected distributions of both classes along either axis become more overlapped and less distinguishable, indicating that the regularization effect of the error tolerance term smooths the feature distributions and mitigates the influence of any dominant direction. This provides further evidence that increasing $\epsilon^2$ improves robustness by reducing sensitivity to label noise.

This example demonstrates that adding the error tolerance term $\frac{\epsilon^2}{d} \boldsymbol{I}$ to the covariance matrix regularizes the feature distributions by smoothing the eigenvalue spectrum and mitigating the dominance of any single principal direction. As a result, the decision boundaries become more robust and less sensitive to small perturbations or label noise. An experimental sensitivity analysis of $\epsilon^2$ in FedCova is provided in Section D.1.

## C  DETAILS OF MAIN EXPERIMENTS

### C.1  EXPERIMENTAL SETUPS

Overall, to evaluate the fundamental learning robustness against noisy labels, we expose the FL system under a more noise-concentrated scenario.

**Federated System.**   To provide a general assessment of robustness, we evaluate the FL system under a noise-sensitive setting. Consider a federated learning system comprising $M = 20$ devices collaboratively training a model. Set one local epoch and full device participation as a general case. Note that some works of FL under noisy labels Xu et al. (2022); Yang et al. (2022); Wu et al. (2023b) consider $M = 100$ devices with a participation fraction set to be 0.1, which, according to their proposed methods, indirectly circumvents those noisy devices by only selecting 10 devices that are detected to be clean.

**Training Task.**   We evaluate the performance of our algorithm on the CIFAR-10, CIFAR-100 Krizhevsky et al. (2009), and a real-world noisy dataset, Clothing1M Xiao et al. (2015). For the network architecture, we employ Resnet-18 He et al. (2016) for CIFAR-10, Resnet-34 for CIFAR-100, and pretrained Resnet-50 for Clothing1M. To obtain the feature representations, we replace the

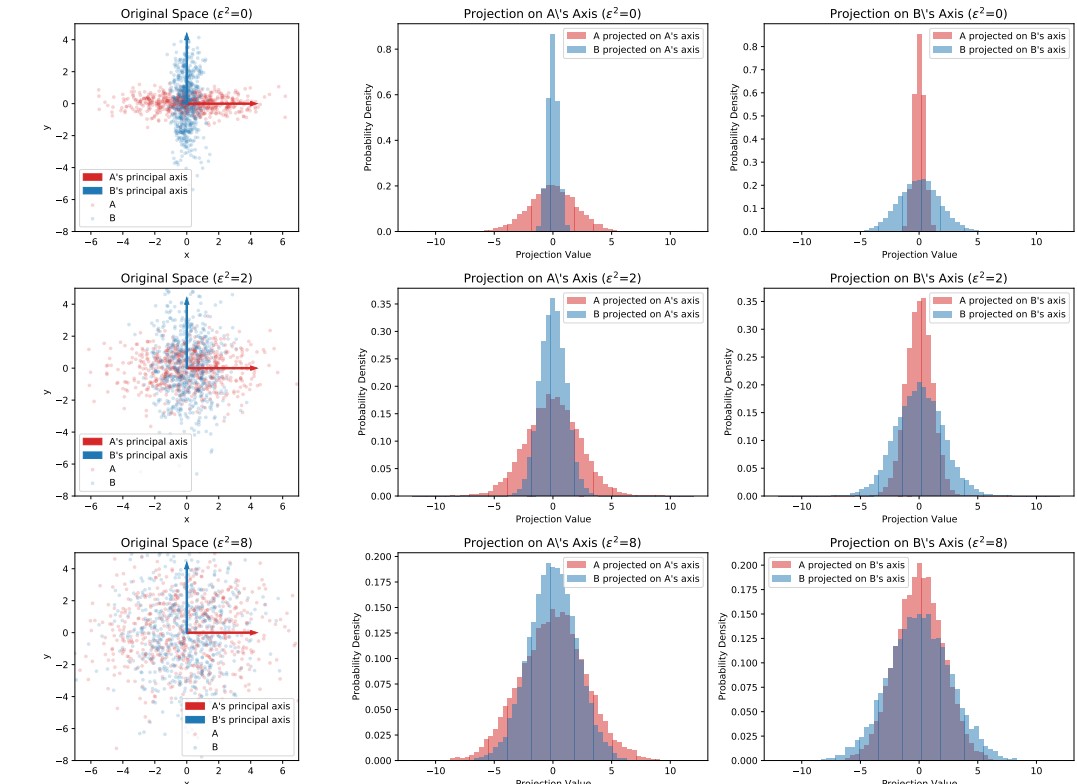

Figure 3: Effect of $\epsilon^2$ illustrated as sample distribution and axis projections.

original final fully connected layer with two fully connected layers, allowing for a transition into the desired feature dimension.

**Data Heterogeneity.** We adopt a challenging data heterogeneity setting in which data distributions across devices are non-i.i.d. Specifically, following a convention setting Xu et al. (2022), we utilize a general partitioning scheme that accounts for both class heterogeneity and dataset size heterogeneity. To create non-i.i.d. partitions, we first construct an indicator matrix $\Psi$ of size $M \times J$, where each entry $\Psi_{m,j}$ indicates whether class $j$ is included in the local dataset of device $m$. Each $\Psi_{m,j}$ is sampled from a Bernoulli distribution with a fixed probability $p$. For each class $j$, we define $\kappa_j$ as the total number of devices that contain samples from class $j$, which is calculated as the sum of the $j$-th column of $\Psi$: $\kappa_j = \sum_{m=1}^{M} \Psi_{m,j}$. Next, for each class $j$, we sample a vector $\boldsymbol{q}_j$ of length $\kappa_j$ from a symmetric Dirichlet distribution parameterized by $\alpha_{\text{Dir}} > 0$. The samples of class $j$ are then randomly allocated to the $\kappa_j$ devices according to the probability distribution $\boldsymbol{q}_j$. This partitioning approach provides a flexible mechanism for controlling both the class distribution among devices and the variation in their local dataset sizes. We set $p = 0.5$ and $\alpha_{\text{dir}} = 5$ as a relatively high non-i.i.d. scenario in the main experiments. The data distribution is illustrated in Fig. 4. We adjust different settings for additional experiments in Section D.4.

**Noise Setting.** To simulate realistic noise scenarios in federated systems, we adopt a bilevel noise model parameterized by the noisy device ratio $\rho$ and the sample noise ratio $\tau$. The noisy device ratio $\rho$ determines the proportion of devices that contain noisy data, while the sample noise ratio $\tau$ controls the fraction of corrupted samples within each noisy device. For each noisy device, its actual sample noise ratio is uniformly sampled from $[0, 2\tau]$ when $\tau \leq 0.5$, or from $[1-2\tau, 1]$ when $\tau \geq 0.5$, ensuring that the average sample noise level is approximately centered around $\tau$. This design allows us to simulate diverse and challenging noise conditions for robustness evaluation. We introduce comprehensive noise patterns from symmetric noise to asymmetric noise following Song & et al. (2022). In the symmetric noise pattern, the mislabeling of a class is corrupted to be a random class, while the flipping of one

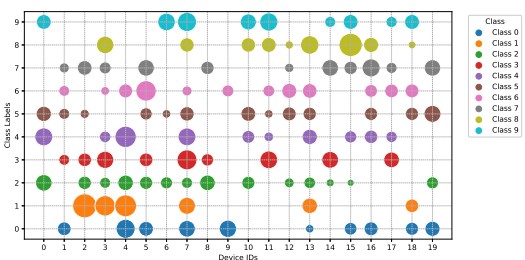

Figure 4: Non-i.i.d. data distribution among devices.

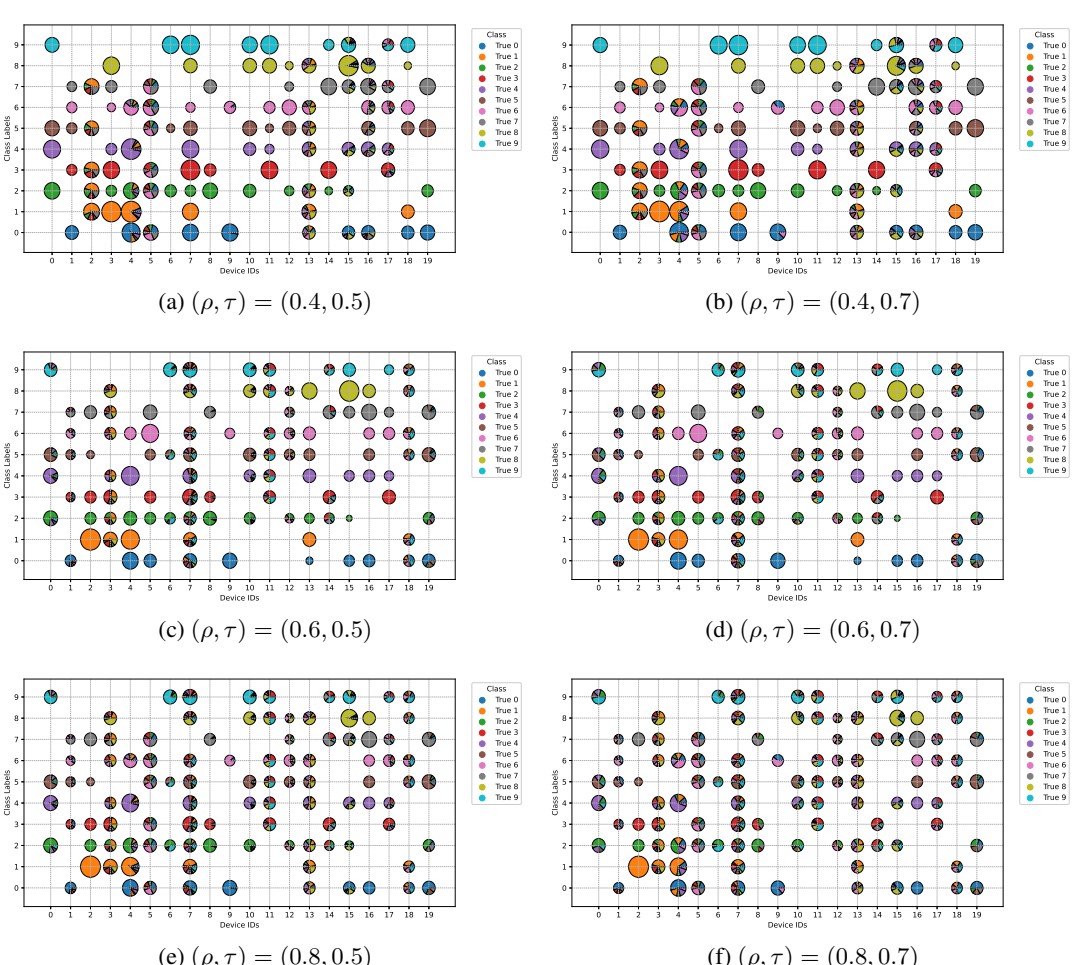

(a) $(\rho, \tau) = (0.4, 0.5)$

(b) $(\rho, \tau) = (0.4, 0.7)$

(c) $(\rho, \tau) = (0.6, 0.5)$

(d) $(\rho, \tau) = (0.6, 0.7)$

(e) $(\rho, \tau) = (0.8, 0.5)$

(f) $(\rho, \tau) = (0.8, 0.7)$

Figure 5: Visualization of different bilevel noise settings.

class is mainly to another specific class in the asymmetric noise pattern. In our main experiments, we adopt noise pairs $(\rho, \tau)$ from $\{(0.4, 0.5),\ (0.4, 0.7),\ (0.6, 0.5),\ (0.6, 0.7),\ (0.8, 0.5),\ (0.8, 0.7)\}$ in symmetric noise pattern and $\{(0.4, 0.3),\ (0.4, 0.5),\ (0.4, 0.7),\ (0.6, 0.3),\ (0.6, 0.5),\ (0.6, 0.7)\}$ in asymmetric noise pattern. The distribution of noisy data across devices and class labels under in symmetric noise pattern is depicted in Fig. 5.

**Parameters.**  We conduct $T = 500$ training rounds in total for Resnet-18 and Resnet-34 and $T = 50$ for pretrained Resnet-50. The learning rate is set as $\eta = 0.1$. The momentum parameter is set as $\beta = 0.9$. The weight decay is set as $\gamma = 5e^{-4}$. We do not use any data mixup augmentation techniques that are part of some baselines' design. We enhance the subspace alignment across

different devices by adding the momentum of the global covariance in the server before broadcasting to devices. We set the feature dimension $d = 128$. For correction, we set the correction threshold as $\eta_c = 0.5$. The correction rounds $\{T_c\}$ are scheduled to start at $T = 200$, with subsequent corrections performed every $\delta = 30$ rounds. In each correction round, correction is performed on the top-$k_1$ noisy devices, while correction with external removal is applied to the top-$k_2$ noisy devices, with $k_1 = 10$ and $k_2 = 5$.

**Computing Resources.** Our experiments are conducted on a GPU server equipped with one NVIDIA A100, one NVIDIA RTX A6000, and one NVIDIA RTX 3090 GPU.

**Baselines.** We evaluate the proposed algorithms compared with several baselines. The main underlying ideas of them, with some specific parameter adjustments, are provided as follows. Unless otherwise specified, the default optimal parameters provided by each method are used. General FL environment settings—including data heterogeneity, label noise configurations, and so on—are not reiterated here; these settings follow those described above and are consistently simulated under the same random seed to ensure fair comparison.

- **FedAvg** McMahan et al. (2017): FedAvg is the classical FL framework for aggregating distributed local models. During each aggregation period, clients upload their model updates to the server, which performs weighted averaging to generate the global model. There is no label correction in FedAvg, which reflects the original overfitting problem for FL under noisy labels.

- **RoFL** Yang et al. (2022): RoFL addresses noisy labels in federated learning by enabling cooperation between the server and local models through the interchange of class-wise feature centroids (means). The server aligns these centroids, which represent central features of local data for each device, and broadcasts the aligned centroids to selected clients in each communication round. This process helps form consistent class decision boundaries among local models, even when noise distributions differ across clients. RoFL also introduces a sample selection approach to filter out data with noisy labels and a label correction method to adjust noisy instances.

- **FedCorr** Xu et al. (2022): One of the most popular frameworks to tackle noisy labels. It classifies devices into clean or noisy categories by evaluating their local intrinsic dimensionality. Furthermore, it employs locally trained filters to distinguish clean samples from noisy devices by examining the training losses of individual samples. For robustness, it employs a proximal term in the loss function and utilizes data augmentation techniques. Given that the total number of rounds in our experiments is $T = 500$, we set the three stages to $T_1 = 5$, $T_2 = 250$, and $T_3 = 250$. The exact communication cost of FedCorr is therefore $20T_1 + T_2 + T_3 = 600$ rounds. We treat the first stage as an additional warm-up phase, even though it incurs a substantial communication overhead.

- **FedNoRo** Wu et al. (2023b): FedNoRo is a two-stage framework designed for noise-robust federated learning under class imbalance and heterogeneous label noise. In the first stage, per-class loss indicators combined with a Gaussian Mixture Model are used for noisy client identification. In the second stage, knowledge distillation and a distance-aware aggregation function are jointly adopted to enable robust federated model updating, addressing both class imbalance and label noise heterogeneity. The warm-up stage is set with $T_1 = 10$ rounds.

- **FedNed** Lu et al. (2024): FedNed addresses extremely noisy clients via a novel negative distillation strategy, where updates from identified noisy clients are leveraged, rather than discarded, to guide the global model away from incorrect directions. This process combines pseudo-labeling and KL-divergence-based loss to enhance robustness against severe label noise. As an exception to all baselines, we set the participation ratio to $0.5$ for FedNed in our experiments to ensure the presence of extremely noisy devices. Otherwise, if all devices participate, we observe that the method fails to distinguish extremely noisy devices.

- **FedNed-** Lu et al. (2024): FedNed- is a degraded variant of FedNed that removes the reliance on extremely noisy devices, making it fully compatible with our setting of severe label noise in federated learning. In this configuration, the method can not distinguish extremely noisy labels and can only leverage the public clean dataset for noise resilience.

- **CoteachingFL**: CoteachingFL adapts the Co-teaching Han et al. (2018) paradigm for federated learning to handle noisy labels. In Co-teaching, two neural networks are trained simultaneously, where each selects a subset of small-loss instances from mini-batches as clean samples and uses them to update its peer network, reducing error accumulation through mutual teaching. This process is executed locally on each client to combat noisy labels, while the two local models are aggregated globally using the FedAvg framework.

- **DivideMixFL**: DivideMixFL adapts the DivideMix Li et al. (2020) framework to federated learning for robust training under noisy labels. DivideMix dynamically divides training data into a labeled (clean) set and an unlabeled (noisy) set using a Gaussian Mixture Model fitted to per-sample loss distributions, then employs semi-supervised learning techniques with two diverged networks for co-division and label co-refinement. Similarly, DivideMix is also performed locally on each client, with the resulting models aggregated globally via FedAvg.

### C.2 LEARNING CURVES

We plot the accuracy curves versus communication rounds of two noise settings with baselines in Fig. 6. We observe clear distinctions among the evaluated methods in terms of robustness to noisy labels. Across both settings, FedCova consistently achieves the highest global accuracy and demonstrates remarkable stability throughout the training process, significantly outperforming all baselines. In particular, FedCova not only converges faster but also maintains a smoother and more robust accuracy trajectory, especially in the presence of increased noise, as seen in Fig. 6b. Compared to traditional aggregation methods such as FedAvg, which suffers from substantial fluctuations and much lower final accuracy, FedCova exhibits improved resilience to the effects of noise and communication variability. Other baselines, including FedCorr, FedNed, and FedNoRo, show competitive performance in the early communication rounds but are eventually surpassed by FedCova as training progresses. Overall, these results highlight the superiority of FedCova in terms of both accuracy and robustness, demonstrating its effectiveness in challenging FL environments with noisy labels under heterogeneous data distributions.

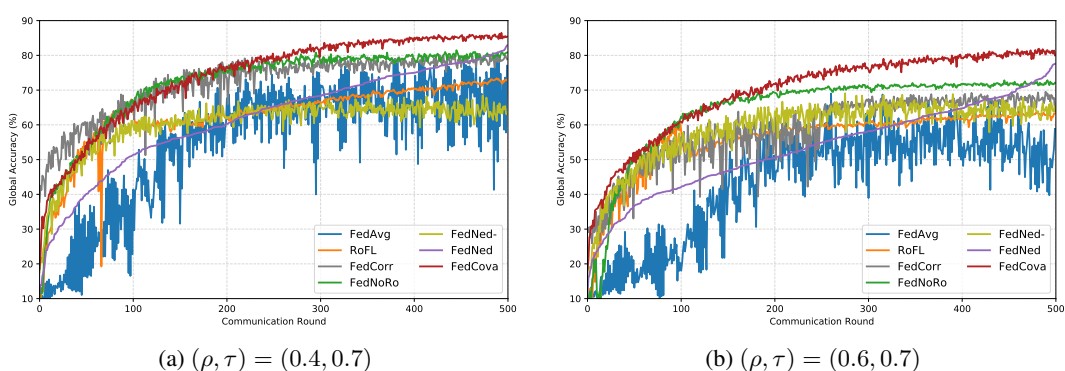

(a) $(\rho, \tau) = (0.4, 0.7)$        (b) $(\rho, \tau) = (0.6, 0.7)$

Figure 6: Accuracy curves versus communication rounds over baselines.

### C.3 SUBSPACE ORTHOGONALITY

To quantitatively assess the improvement of the orthogonality among different class covariance principal directions during training, we compute the mean absolute cosine similarity between the principal eigenvectors of the class covariance matrices. Specifically, for each class $j$ with covariance matrix $\mathbf{\Sigma}_j$, we extract its principal eigenvector $\mathbf{v}_j$, corresponding to the largest eigenvalue. We then construct the matrix $\mathbf{V} = [\mathbf{v}_1, \mathbf{v}_2, \ldots, \mathbf{v}_J]^\top \in \mathbb{R}^{J \times d}$, where $J$ is the number of classes and $d$ is the feature dimension. The orthogonality matrix $A$ is defined as $A_{uv} = |\mathbf{v}_u^\top \mathbf{v}_v|$, representing the absolute value of the cosine similarity between the principal directions of classes $u$ and $v$. We define

a measurement, orthogonality index $\phi$, as

$$\phi = \frac{1}{J(J-1)} \sum_{u \neq v} |\mathbf{v}_u^\top \mathbf{v}_v|,$$  (31)

where a lower value of $\phi$ indicates that the principal directions of the class covariances are more orthogonal, reflecting greater diversity and discriminability among classes. Conversely, a higher value of $\phi$ suggests increased alignment of the principal directions, implying reduced separability between classes.

Fig. 11 illustrates the evolution of both the orthogonality index $\phi$ and accuracy as a function of communication rounds under two noise settings. As training progresses, $\phi$ exhibits a rapid decline, approaching a low and stable value after approximately 100 rounds. This trend indicates that the principal directions of the class covariance matrices become increasingly orthogonal under the training of our objective function, suggesting enhanced inter-class discrimination. Notably, the reduction in $\phi$ coincides with a marked increase in global accuracy, as depicted on the right axis. This correlation implies that the model's ability to learn more orthogonal (i.e., more discriminative) class directions directly contributes to performance gains. Furthermore, under the higher noise setting $(\rho, \tau) = (0.6, 0.7)$, $\phi$ decreases more slowly compared to the lower noise case, reflecting the impact of noise on feature alignment and the enhanced tolerance provided by the error tolerance term. Nevertheless, the overall trend remains consistent across both settings, underscoring the robustness of the proposed approach in promoting class separability and achieving superior accuracy even in severe label noise conditions.

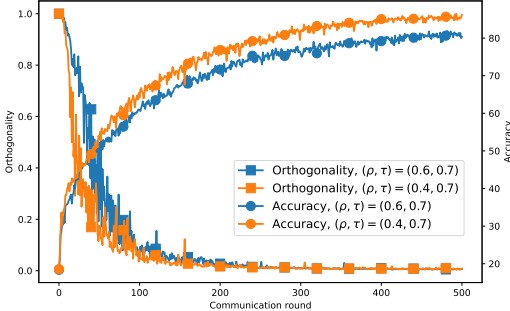

Figure 7: Orthogonality and accuracy curves versus communication rounds.

## C.4 Correction Performance

We provide a comprehensive evaluation of the correction performance of FedCova under different label noise settings in Figs. 8 and 9. As illustrated in Fig. 8, introducing the correction leads to a continuous and substantial reduction in the number of total noisy samples as communication rounds progress. For both noise settings, $(\rho, \tau) = (0.4, 0.7)$ and $(\rho, \tau) = (0.6, 0.7)$, the global noise rate exhibits a clear downward trajectory, which is accompanied by a steady improvement in global test accuracy. Specifically, under $(\rho, \tau) = (0.4, 0.7)$, the global noise rate drops from 27.51% to 6.93%; for the more challenging case $(\rho, \tau) = (0.6, 0.7)$, it decreases from 34.42% to 11.20%. Notably, even in the scenario with higher initial noise, the correction mechanism remains highly effective in driving down noisy samples and enhancing accuracy, demonstrating robustness to severe label corruption.

Fig. 9 further breaks down the correction performance at the device level for both noise settings. The number of noisy samples before and after correction is shown for each device. It is evident that the correction procedure achieves significant noise reduction across devices, with the number of noisy samples after correction (purple bars) being substantially lower than before correction (green bars). This effect is consistent whether the devices have large or small local datasets and have large or small sample noise ratios on devices, indicating that the correction strategy is effective across heterogeneous local data distributions.

Overall, these results highlight the efficacy and robustness of our correction mechanism. The global reduction in noise and improvement in accuracy, together with the consistent device-level denoising

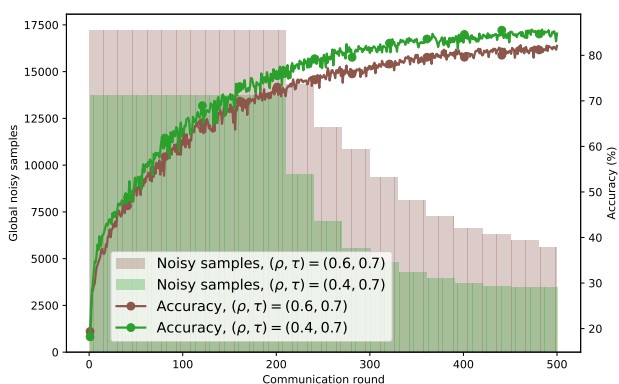

Figure 8: Correction performance and accuracy versus communication rounds.

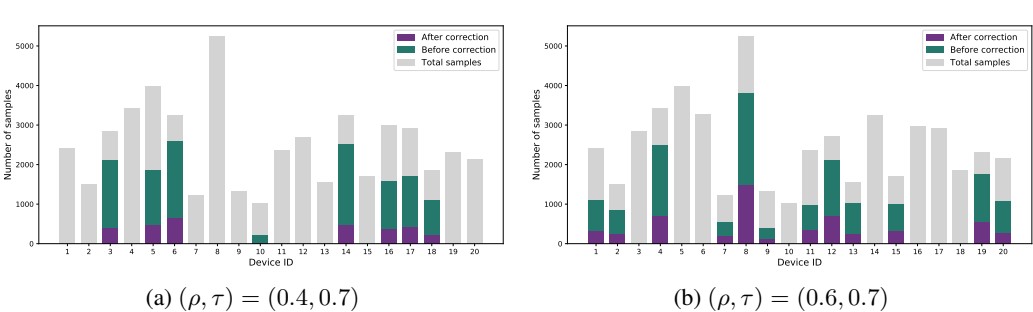

(a) $(\rho, \tau) = (0.4, 0.7)$            (b) $(\rho, \tau) = (0.6, 0.7)$

Figure 9: Correction performance on devices.

observed across varying degrees of heterogeneity and noise, demonstrate that our periodic correction mechanism, assisted by the precise classifier and external corrector, can effectively mitigate the impact of label noise by relabeling datasets.

# D  ADDITIONAL EXPERIMENTAL RESULTS

## D.1  EFFECT OF ERROR TOLERANCE $\epsilon^2$

Fig. 10 systematically investigates the impact of the error tolerance parameter $\epsilon^2$, which is introduced as an isotropic regularization term $\frac{\epsilon^2}{d} \boldsymbol{I}$ to each class covariance matrix. The experiment is conducted under the noise setting $(\rho, \tau) = (0.6, 0.7)$ with other parameters set as default. This regularization serves to smooth the eigenvalue spectrum of the covariance, mitigating the dominance of any single principal direction and effectively relaxing strict orthogonality constraints among class-specific feature subspaces. Such a mechanism is designed to enhance robustness against noisy or mislabeled samples by preventing the loss function from being overly sensitive to small perturbations in the data.

As shown in Fig. 10a, increasing $\epsilon^2$ from 0.5 to 2 leads to a significant improvement in both convergence speed and final accuracy, as the added regularization enables the model to better tolerate label noise and outliers. This is further confirmed by Fig. 10b, where the achievable accuracy sharply increases with $\epsilon^2$ in this range, peaking around $\epsilon^2 = 6$. This demonstrates that moderate regularization allows the model to focus on the essential statistical structure of the feature space, rather than overfitting to specific feature values or noisy labels. However, as $\epsilon^2$ increases beyond the optimal range, the accuracy begins to decline. Specifically, for larger values such as $\epsilon^2 = 20$ and $\epsilon^2 = 30$, a clear drop in achievable accuracy is observed. This can be attributed to excessive smoothing, which overly relaxes the class boundaries and diminishes the discriminative capacity of the learned feature representations. In other words, while introducing $\epsilon^2$ is crucial for promoting

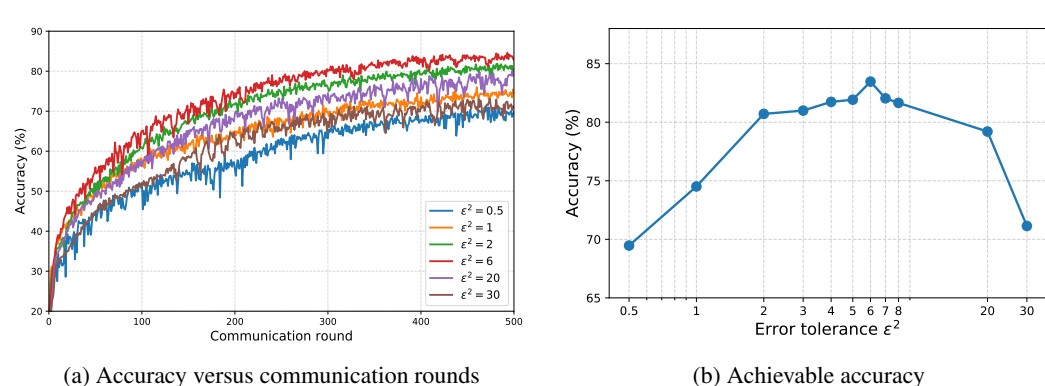

(a) Accuracy versus communication rounds

(b) Achievable accuracy

Figure 10: Accuracy under different error tolerance $\epsilon^2$.

robustness and stabilizing training, too large an error tolerance compromises class separability by making the feature subspaces less distinctive.

Overall, these results highlight the dual role of the error tolerance parameter: it provides essential robustness to noise and outliers by regularizing the covariance spectrum, but it requires careful tuning to balance model flexibility and discriminability. The observed accuracy trends validate the theoretical motivation for including the $\frac{\epsilon^2}{d}\boldsymbol{I}$ term.

**Discussions on $\epsilon$ from the perspective of privacy.** As $n \sim \mathcal{N}\left(\boldsymbol{0}, \frac{\epsilon^2}{d}\boldsymbol{I}\right)$ is added as the Gaussian noise in Eq. (5), it aligns the noise injection as for Differential Privacy (DP) for privacy protection. Thus, from the perspective of features, we inherently introduce the corresponding noise. Considering the privacy issue, the trade-off is far more than only over the noise resilience and model discriminability, but also the performance and the privacy protection level. With the increase of the $\epsilon$, higher privacy is guaranteed, but accuracy faces further challenge as features are further noised.

## D.2 EFFECT OF AUGMENTATION COEFFICIENT $\alpha$

In this section, we investigate how different settings of the augmentation coefficient $\alpha$ in the subspace-augmented classifier affect model performance under label noise. The experiment is conducted under the noise setting $(\rho, \tau) = (0.6, 0.7)$ with all other hyperparameters set to their default values. Recall that the augmentation coefficient $\alpha$ in the subspace-augmented classifier controls the exponent in the generalized classifier, as formulated in Eq. equation 11. Specifically, $\alpha = 1$ recovers the standard Mahalanobis distance used in the MAP classifier, while $\alpha = 2$ corresponds to the projection classifier proposed in Chan et al. (2022). Larger values of $\alpha$ in general enhance the discriminative power of the classifier by amplifying differences across class-specific feature subspaces, but may also reduce tolerance to label noise and outliers.

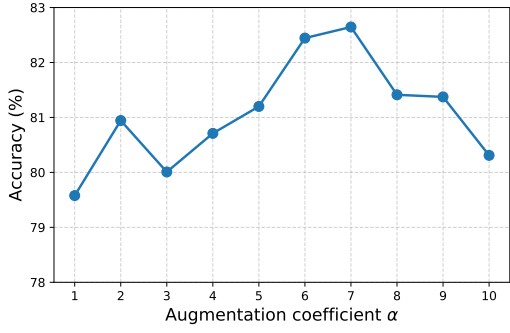

Figure 11: Accuracy under different augmentation coefficient $\alpha$.

As shown in Fig. 11, the test accuracy exhibits a non-monotonic dependence on $\alpha$. Generally, accuracy increases as $\alpha$ rises from 1 to 8, highlighting the effectiveness of subspace augmentation for improving discrimination under noisy label conditions. The best performance is achieved at $\alpha = 7$. The special performance in $\alpha = 2$ (the setting used in the main experiments) may be attributed to the fact that the default hyperparameters, including $\epsilon^2$, are tuned under this setting, suggesting that the error tolerance and discriminative strength need to be co-adapted for optimal results. Beyond $\alpha = 7$, further increasing $\alpha$ induces the accuracy to start declining. This trend indicates that while increasing $\alpha$ initially strengthens class separation by penalizing deviations along minor axes of the covariance, an excessively large $\alpha$ may cause the classifier to become overly sensitive to small perturbations and estimation noise in the covariance, thus hindering robustness under noisy labels.

These results demonstrate the effectiveness of our subspace augmentation strategy: introducing the augmentation coefficient $\alpha$ enhances model discrimination and leads to significant performance gains under label noise. Moreover, an appropriate coordination between the discriminative power brought by augmentation and the relaxation effect induced by error tolerance can be beneficial for achieving balanced performance against label noise.

### D.3 EFFECT OF FEATURE DIMENSION

We analyze how the choice of feature dimension $d$ influences the trade-off between discriminative power and robustness to noisy labels in our covariance-based feature learning framework. The experiment adopts the noise configuration $(\rho, \tau) = (0.6, 0.7)$, while all other hyperparameters remain at their default settings.

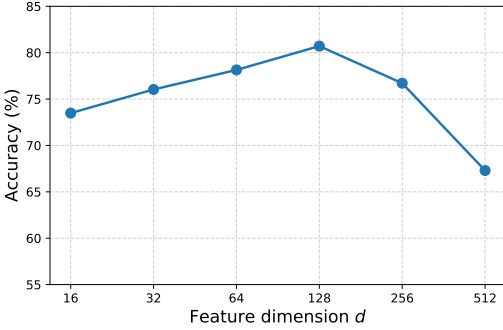

Figure 12: Accuracy under different feature dimension $d$.

As illustrated in Fig. 12, the classification accuracy first increases with $d$, reaching its peak at $d = 128$, and then declines as $d$ continues to grow. This trend highlights two key phenomena. On the one hand, increasing the feature dimension provides a richer representation space, allowing the covariance matrices to capture more nuanced intra-class and inter-class variations, which enhances the discriminative capability of the model. On the other hand, excessively large feature dimensions may introduce over-parameterization, making the covariance estimation less reliable–especially under the presence of noisy labels and limited sample sizes per class. Achieving alignment of principal directions across different devices also becomes increasingly challenging. In such cases, the model can become more susceptible to noise, leading to a degradation in accuracy.

**Computational Complexity and Storage Overhead.** To assess the practical deployability of FedCova, we analyze its computational and storage complexity. As $B$ denotes the batch size, $d$ the feature dimension, and $J$ the number of classes. The backbone network (e.g., ResNet-18) dominates the computational cost with $\mathcal{O}(10^9)$ FLOPs per iteration. FedCova introduces two additional steps: (1) Covariance Accumulation: Computing the covariance matrix requires $\mathcal{O}(Bd^2)$. (2) Loss Computation: Calculating $\log \det(\boldsymbol{\Sigma})$ and its gradients requires $\mathcal{O}(d^3)$ via Cholesky decomposition. With $d = 128$, the $\mathcal{O}(d^3)$ term corresponds to $\approx 2$ MFLOPs, which is three orders of magnitude smaller than the backbone computation. Furthermore, these matrix operations are naturally parallelizable on GPUs using standard libraries (e.g., cuBLAS), ensuring they do not become a bottleneck. For the storage overhead. The client must store $J$ covariance matrices of size $d \times d$. The storage complexity is

$\mathcal{O}(Jd^2)$. For $J = 10$ and $d = 128$, the storage requirement for covariance matrices is approximately 0.6 MB (using 32-bit floats). This is trivial for modern mobile devices compared to the memory footprint of the model itself (e.g., $\approx 45$ MB for ResNet-18). For a dataset with $J = 100$ classes, the total memory overhead is approximately 6 MB (compared to $\approx 83$ MB for ResNet-34). This is well within the capacity of resource-constrained mobile devices.

**Communication Overhead Analysis.** The feature covariance matrices communicated in our framework have size $d \times d$ for each of the $J$ classes. Taking the setting used in our main experiments where $d = 128$ and $J = 10$, the total covariance parameters to be transmitted are $J \times d^2 = 10 \times 128^2 = 163{,}840$. In comparison, the number of parameters in a standard ResNet-18 model is approximately 11.2 million. Therefore, the additional communication overhead introduced by transmitting all class covariance matrices amounts to only about $1.4\%$ of the model parameters, which is two orders of magnitude smaller. This demonstrates that the additional communication cost is negligible relative to the overall model size in practical federated learning scenarios.

Taken together, these results highlight the necessity of carefully choosing the feature dimension $d$ to achieve a balance between classification discriminability, robustness to noise, and communication efficiency.

### D.4 DIFFERENT DATA HETEROGENEITY SETTINGS

We further investigate the performance of FedCova under different data heterogeneity by varying the non-i.i.d. partitioning parameters $(p, \alpha_{\text{dir}})$. As described in the technical section, $p$ controls the probability that each device contains a particular class, while $\alpha_{\text{dir}}$ determines the degree of sample size variation across devices for each class. This flexible partitioning scheme allows us to simulate different levels of heterogeneity in both class distribution and local dataset sizes.

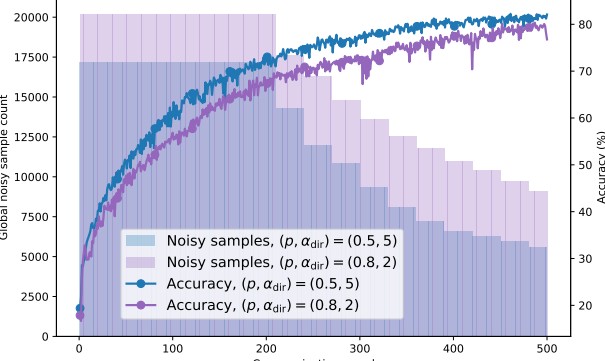

Figure 13: Correction performance and accuracy under different data heterogeneity settings.

Fig. 13 presents the correction performance and accuracy under two representative heterogeneity settings: $(p, \alpha_{\text{dir}}) = (0.5, 5)$ (higher class diversity per device, more balanced sizes) and $(p, \alpha_{\text{dir}}) = (0.8, 2)$ (lower class diversity per device, more skewed sizes). The experiment is conducted under the noise setting $(\rho, \tau) = (0.6, 0.7)$, with all other hyperparameters set to their default values. As shown in Fig. 13, our correction framework consistently reduces the global noise rate over training. Specifically, under $(p, \alpha_{\text{dir}}) = (0.5, 5)$, the global noise rate drops from 34.42% to 11.20%, while for $(p, \alpha_{\text{dir}}) = (0.8, 2)$, it decreases from 40.33% to 18.16%. This demonstrates the effectiveness of our periodic correction mechanism even in more challenging heterogeneous settings. Even under the more challenging heterogeneity setting of $(p, \alpha_{\text{dir}}) = (0.8, 2)$, our method still achieves notable reductions in noise rate and maintains considerable accuracy, demonstrating its robustness to severe data heterogeneity.

## E    LIMITATIONS AND FUTURE WORK

Despite the effectiveness of our proposed FedCova framework in achieving robustness against noisy labels within a covariance-aware FL paradigm, it introduces the transmission of covariance matrices. Future work may explore privacy-preserving analysis and communication-efficient strategies for handling covariances. For instance, one could consider incorporating differential privacy mechanisms for covariance matrices, or further reduce communication overhead by transmitting only the most salient principal components. Such enhancements could improve the scalability and practicality of our approach in large-scale federated learning deployments.

## F    STATEMENT ON THE USE OF LARGE LANGUAGE MODELS (LLMs)

We disclose that the LLMs were only used to polish the writing and grammar in this paper.

