# OpenReview forum: "FedCova: Robust Federated Covariance Learning Against Noisy Labels"
_ICLR.cc/2026/Conference — Submitted to ICLR 2026_

### Official Review · Reviewer_avwp · 2025-10-27

**Soundness:** 2
**Presentation:** 2
**Contribution:** 2
**Rating:** 4
**Confidence:** 2

**Summary:**

This paper introduces FedCova, a federated learning framework designed to address the challenge of noisy labels in distributed datasets. The framework integrates three components through a unified covariance-based approach: (1) lossy feature encoding via mutual information maximization with an error tolerance term, (2) intrinsic classifier construction using covariance-based MAP estimation with subspace augmentation, and (3) label correction using external correctors to avoid self-bias.

**Strengths:**

1. The main idea is interesting. The paper presents a principled information-theoretic approach by maximizing mutual information I(Z;Y) while focusing on covariance structures rather than mean statistics.
2.  The integration of feature encoding, classification, and label correction through covariance structures provides an elegant and cohesive solution, avoiding the need for auxiliary clean datasets or duplicate models required by existing methods.
3. The covariance transmission overhead is only ~1.4% of model parameters, making the approach practical for federated settings.

**Weaknesses:**

1. While the mutual information objective is well-motivated, the paper lacks formal convergence guarantees or theoretical bounds on the robustness to label noise.
2. The method introduces several hyperparameters (ε², α, d, ηc) that require tuning.
3. The approach requires estimating and storing J×d×d covariance matrices. For problems with many classes (J>>100) or high feature dimensions, this could become prohibitive despite the claimed efficiency.
4. Only symmetric noise is considered; asymmetric or instance-dependent noise patterns are not evaluated. And the largest dataset (Clothing1M) still has relatively few classes (14)
5. There is no comparison with recent self-supervised or semi-supervised federated learning methods.

**Questions:**

1. How does the method perform under class imbalance, where some classes have significantly fewer samples?
2. Can you provide theoretical analysis on the convergence rate or sample complexity bounds?
3. How sensitive is the method to the correction schedule {Tc}?
4. Could the framework be extended to handle feature noise in addition to label noise?

---

> ### Author Response · Authors · 2025-11-25
> **Response to Official Review by Reviewer avwp (1/4)**
>
> *We sincerely appreciate the reviewer for recognizing our contributions,*  ***especially the interesting idea of covariance-ware robustness design with information-theoretical analysis, the unified design from learning to correction via covariance structure, and the efficiency discussion.***
>
> *Meanwhile, we thank the reviewer for the constructive comments. Our point-to-point responses to concerns on Weaknesses and Questions are given below.*
>
> ### **<Reply to Weakness 1>**
>
> We appreciate the reviewer's rigorous inquiry. We understand the concern arises because FedCova dynamically updates the labels, making the optimization objective technically non-stationary. However, we respectfully clarify that the convergence of our framework does not require analyzing the "convergence of input data" in the traditional sense. Instead, it relies on the **asymptotic stability of the correction mechanism**, which ensures the system reduces to a standard optimization problem.
>
> **(1) The decoupling of optimizer and data dynamics.** Standard convergence analysis focuses on the relationship between the loss function $L(\theta)$ and parameters $\theta$. In FedCova, although the dataset $\mathcal{D}^t$ evolves, the optimization step $\theta^{t+1} \leftarrow \theta^t - \eta \nabla L(\theta^t; \mathcal{D}^t)$ strictly follows the Gradient Descent rule. The optimizer itself retains its descent properties at every step, provided the objective function does not oscillate violently. In other words, FedCova can be regarded as a “plug-and-play” framework compatible with standard FL optimizers. To make it clear, we have polished the overview figure to better illustrate the proposed framework. The revised figure is available at: https://github.com/anonymouslinkforrebuttal/iclr2025_12237/blob/main/modelscheme.png. As in Fig. 1, FedCova (core steps as the circle of red arrows) functions as a protective "fortress" wrapping around the conventional FL process (the circle of blue arrows), rather than fundamentally altering the underlying gradient descent optimization backbone. Consequently, the convergence properties of FedCova inherently align with those of the standard FL algorithms (e.g., FedAvg).
>
> **(2) Stability of label correction.** The core reason FedCova converges is that the label correction is not stochastic but driven by the intrinsic classifier based on the structure of the feature space (specifically, the covariance statistics).  As the feature extractor improves, the class-specific covariance matrices $\boldsymbol{\Sigma}_j$ become more accurate estimates for class discrimination. Consequently, the corrected labels $\hat{y}$ rapidly stabilize. Once the label assignments stop flipping (i.e., $\mathcal{D}^t \approx \mathcal{D}^{t+1}$), the training process becomes mathematically equivalent to standard FedAvg on a fixed, updated dataset. Therefore, our method inherits the standard convergence guarantees of standard FL. This theoretical stability is **numerically verified** in Fig. 8 in Appendix C.4, which is available in [https://github.com/anonymouslinkforrebuttal/iclr2025_12237/blob/main/correction.png](https://github.com/anonymouslinkforrebuttal/iclr2025_12237/blob/main/modelscheme.png), where the training converges monotonically and smoothly. This confirms that the dynamic correction serves as a progressive label denoising improvement rather than a source of divergence.
>
> ### **<Reply to Weakness 2>**
>
> Thanks for the comment.
>
> **(1) Hyperparameters $d$ and $\eta_c$ are not required to be tuned.** In applications, they can be set just by following the paper as $d=128$ and $\eta_c=0.5$ for the default settings. Thus, FedCova introduces only two key hyperparameters that require tuning: $\epsilon$ and $\alpha$.
>
> **(2)** **The introduction of additional hyperparameters is minor and acceptable** **compared with state-of-the-art baselines.** As summarized in the table below, our parameter overhead is comparable to or lower than other methods. Crucially, unlike methods such as RoFL, which involve multiple coupled parameters (e.g., balancing two $\lambda$s), our parameters are independent and work in different steps and are easier to manage.
>
> |Method|RoFL|FedCorr|FedNoRo|FedNed-|FedNed|**FedCova**|
> |-|-|-|-|-|-|-|
> |**Key Hyperparameters**|$T_p, \lambda_{cen}, \lambda_e$|$T_1, T_2, T_3, \theta, \pi$|$\lambda, T_1$|$T_0$|$T_0, \lambda$|$\epsilon, \alpha$|
> |**Sensitivity Analysis Provided?**|No|No|Yes|No|Yes|**Yes**|

---

> ### Author Response · Authors · 2025-11-25
> **Response to Official Review by Reviewer avwp (2/4)**
>
> *(cont’d)*
>
> **Meanwhile, the key parameters do have principled guidance and are not overly sensitive to tuning based on the sensitivity analysis.** Take $\epsilon$ for instance, it initially has conceptually physical meanings of the injected noise into the feature spaces. As it increases, the noise resilience ability can be enhanced while the class discriminability can be weakened. The balance of it is the interesting point of FedCova as we discussed in the paper. Moreover, we introduce a toy example of the effect of the error tolerance term controlled by $\epsilon$ in Appendix B.3, which is accompanied by a visual illustration of tuning it for balancing noise resilience and class discriminability. Additionally, the sensitivity analysis in Appendix D.1 and D.2 shows that tuning $\epsilon$ and $\alpha$ in an appropriate region is not sensitive (for instance, the accuracy remains around 80–83% as $\epsilon \in [1,10]$ and $\alpha \in [2,9]$), which is convenient for application without tuning. The optimal values also remain consistent across different datasets.
>
>
> ### **<Reply to Weakness 3>**
>
> Thanks for the comment.
>
> **(1) (Micro-level Analysis) Firstly, theoretical computation complexity and storage are analyzed, which indicates the acceptable complexity of our framework.** For computation complexity, Let $B$ be the batch size and $d$ the feature dimension. The covariance calculation is $\mathcal{O}(Bd^2)$ and the matrix operations (determinant/inverse) are $\mathcal{O}(d^3)$. When the ResNet-18 requires $\approx 0.5$ GFLOPs per pass, FedCova's matrix operations require $\approx 2$ MFLOPs with $d=128$, indicating the cost is merely 0.4% of the backbone's. Furthermore, these matrix operations are naturally parallelizable on GPUs using standard libraries (e.g., cuBLAS), ensuring they do not become a bottleneck.
>
> The storage complexity is $\mathcal{O}(Jd^2)$. For a large dataset with $J=100$ classes, the covariance storage is $\approx 6$ MB. This is trivial compared to the model size (e.g., $\approx 83$ MB for ResNet-34) and easily fits on mobile devices.
>
> Accordingly, we have provided detailed discussion in the revised paper.
>
> > [Appendix D.3 Paragraph 3] Computational Complexity and Storage Overhead.
> To assess the practical deployability of FedCova, we analyze its computational and storage complexity.
> As $B$ denotes the batch size, $d$ the feature dimension, and $J$ the number of classes. The backbone network (e.g., ResNet-18) dominates the computational cost with $\mathcal{O}(10^9)$ FLOPs per iteration. FedCova introduces two additional steps:
> (1) Covariance Accumulation: Computing the covariance matrix requires $\mathcal{O}(B d^2)$.
> (2) Loss Computation: Calculating $\log \det(\boldsymbol{\Sigma})$ and its gradients requires $\mathcal{O}(d^3)$ via Cholesky decomposition.
> With $d=128$, the $\mathcal{O}(d^3)$ term corresponds to $\approx 2$ MFLOPs, which is three orders of magnitude smaller than the backbone computation. Furthermore, these matrix operations are naturally parallelizable on GPUs using standard libraries (e.g., cuBLAS), ensuring they do not become a bottleneck.
> >
> >
> > For the storage overhead. The client must store $J$ covariance matrices of size $d \times d$. The storage complexity is $\mathcal{O}(J d^2)$. For $J=10$ and $d=128$, the storage requirement for covariance matrices is approximately $0.6$ MB (using 32-bit floats). This is trivial for modern mobile devices compared to the memory footprint of the model itself (e.g., $\approx 45$ MB for ResNet-18). For a dataset with $J=100$ classes, the total memory overhead is approximately $6$ MB (compared to $\approx 83$ MB for ResNet-34). This is well within the capacity of resource-constrained mobile devices.
> >
>
> **(2) (Macro-level Analysis) The runtime and efficiency compared baselines are added, numerically supports the practicability.** Even though the training speed up is not our primary contribution, we still follow the reviewer’s comment to summarize the experimental runtime for comparison. We measured the total runtime on NVIDIA RTX A6000 GPU. FedCova is only $1.6\times$ the time of standard FedAvg, whereas most other robust methods range from $2.2\times$ to $5.2\times$.
>
> |Method|FedAvg|CoteachingFL|DivideMixFL|RoFL|FedCorr|FedNoRo|FedNed-|FedNed|FedCova|
> |---|---|---|---|---|---|---|---|---|---|
> |Runtime(min)|290|1217|1508|383|643|664|332|390|467|
> |Runtime(Ratio to FedAvg)|1.0x|4.2x|5.2x|1.3x|2.2x|2.3x|1.1x|1.3x|**1.6x**|

---

> ### Author Response · Authors · 2025-11-25
> **Response to Official Review by Reviewer avwp (3/4)**
>
> *(cont’d)*
>
> CoteachingFL and DivideMixFL run the slowest due to simultaneously training two models. Compared to others, FedCova achieved balanced efficiency as it removes the need for external dependencies required by other methods, as shown in the table below. FedCova outperforms most baselines like FedCorr and FedNoRo, primarily because FedCova eliminates the expensive warm-up phases that require hundreds of extra device-server exchanges. Regarding the "No Free Lunch" theorem, while FedCova's runtime is slightly higher than FedNed, this comparison requires context: FedNed relies on a clean public dataset, an impractical condition in real-world scenarios that offers an unfair advantage akin to accessing ground truth. By solving the noise problem internally without relying on clean data or expensive warm-up phases, FedCova offers a superior balance of performance and deployability.
>
> |Method|FedAvg|CoteachingFL|DivideMixFL|RoFL|FedCorr|FedNoRo|FedNed-|FedNed|**FedCova**|
> |-|-|-|-|-|-|-|-|-|-|
> |**Additional Conditions**|-|Double model|Double model|-|Warm-up Rounds|Warm-up Rounds|Clean Dataset|Clean Dataset, Extremely Noisy Devices|**-**|
>
>
> ### **<Reply to Weakness 4>**
>
> **We added the experiments under the asymmetric noise pattern and accidentally found further competitiveness of FedCova.** Thanks for the constructive suggestion. Following the reviewer’s suggestion, we have conducted additional experiments on asymmetric noise patterns with the results summarized in the following table.
>
> |Method|$\rho=0.4,\tau=0.3$|$\rho=0.4,\tau=0.5$|$\rho=0.4,\tau=0.7$|$\rho=0.6,\tau=0.3$|$\rho=0.6,\tau=0.5$|$\rho=0.6,\tau=0.7$|
> |---|---|---|---|---|---|---|
> |FedAvg|$72.06\pm4.35$|$71.23\pm5.86$|$67.40\pm11.2$|$68.91\pm5.02$|$63.53\pm2.89$|$43.40\pm18.2$|
> |RoFL|$78.87\pm0.10$|$77.85\pm0.53$|$71.56\pm0.46$|$78.45\pm0.32$|$66.10\pm2.68$|$41.38\pm1.00$|
> |FedCorr|$75.21\pm0.57$|$62.66\pm1.18$|$38.67\pm1.23$|$39.37\pm0.49$|$34.00\pm0.29$|$30.66\pm0.25$|
> |FedNoRo|$86.40\pm0.72$|$85.72\pm0.41$|$84.76\pm0.62$|$75.27\pm0.21$|$63.88\pm19.90$|$33.42\pm37.8$|
> |FedNed-|$75.69\pm0.98$|$70.38\pm1.40$|$65.51\pm3.48$|$78.63\pm0.37$|$71.22\pm1.15$|$66.44\pm1.44$|
> |FedNed|$82.30\pm0.35$|$79.73\pm0.43$|$78.54\pm0.65$|$79.77\pm0.57$|$78.03\pm0.34$|$74.67\pm0.66$|
> |**FedCova**|$\mathbf{88.00\pm0.25}$|$\mathbf{87.84\pm0.24}$|$\mathbf{87.77\pm0.19}$|$\mathbf{87.83\pm0.40}$|$\mathbf{87.31\pm0.39}$|$\mathbf{87.29\pm0.26}$|
>
> The analysis is also provided in the revised paper as follows.
>
> > [Section 5.2 Paragraph 1] Numerical comparison results in the asymmetric noise pattern are summarized in Table 2. Under such a serious asymmetry, baselines like FedCorr and FedNoRo are more sensitive to high noise levels. In contrast, FedCova demonstrates superior robustness under noisy labels. As the noise level changes from $(0.4, 0.3)$ to $(0.6, 0.7)$, the test accuracy is guaranteed around 87--88\%. This can inherently be attributed to the orthogonality structure of the class subspaces driven by FedCova. When in asymmetric noise, the tangle is over fewer classes, extremely between two classes, the subspaces are then only mixed mainly over two classes, which is more comfortable for the discrimination in FedCova.
> >
>
> In sum, FedCova is designed to be independent of specific noise patterns by enforcing intrinsic feature learning robustness rather than modeling noise transitions. Unlike some methods that rely on estimating a noise transition matrix (as the reviewer worries), FedCova leverages the feature covariance to construct robustness through class-subspace orthogonality. Different types of noise patterns typically manifest as different patterns of class deviations that confuse the decision boundary. Symmetric noise implies a uniform blurring across all class boundaries, whereas asymmetric noise corresponds to blurring between specific class pairs. However, FedCova consistently drives these feature subspaces to be orthogonal. This orthogonality constraint is universally valid and robust regardless of the noise type: FedCova effectively disentangles the features, whether the boundary blurring is global (symmetric) or local (asymmetric).
>
> As for Clothing1M, this is the most popular open-source real-world dataset with noisy labels that is used to validate robust learning algorithms, following many reviewed studies in the related works. The difficulty of it already comes from the real annotation errors from the real-world clothing figures. For evaluation with a larger number of classes, we have conducted on CIFAR-100. We believe that simultaneously including the two experiments can promote comprehensive validation and solve the reviewer’s concerns.

---

> ### Author Response · Authors · 2025-11-25
> **Response to Official Review by Reviewer avwp (4/4)**
>
> ### **<Reply to Weakness 5>**
>
> **Comparison with semi-supervised learning mechanisms has been included.** Thanks for the comment. Actually, we clarify that our comparison already includes state-of-the-art methods that incorporate semi-supervised learning mechanisms, FedNoRo and FedNed. In the context of Noisy Label Learning, treating noisy samples as unlabeled data or correcting them via pseudo-labels is the dominant semi-supervised paradigm. As claimed in their papers, FedNed is a representative semi-supervised framework that employs negative distillation. It identifies noisy clients using uncertainty estimation and, instead of discarding them, utilizes their updates as "bad teachers" while generating **pseudo-labels** to guide the training on noisy data. This explicitly treats noisy samples as unlabeled data for semi-supervised training to improve model reliability. FedNoRo adopts a two-stage semi-supervised framework. In its second stage, it applies knowledge distillation for noisy clients, generating **soft labels** from the global model to supervise local training. This mechanism effectively treats noisy data as partially unlabeled and leverages soft targets for correction. As shown in Table 1-4, **FedCova consistently outperforms these semi-supervised methods (FedNed, FedNoRo)**. Unlike these baselines, which often require complex multi-stage training or heuristic pseudo-labeling, FedCova achieves superior robustness through a unified, covariance-based feature subspace alignment strategy.
>
> ### **<Reply to Question 1>**
>
> Thanks for the comment. The class imbalance problem in federated learning is regarded as the data heterogeneity issue over edge devices. Typical solutions in conventional federated learning focus on a deliberate alignment of the local model parameters (such as FedProx, which is utilized in one of the baselines in our paper, FedCorr [50]) or the auxiliary alignment of the means of the features (such as FedProto [40], a similar solution is RoFL [52] in our baselines). As the reviewer's concerns, the problem can be further terrible in the noisy setting, as the model is less confident and the means are sensitive to label noise. **In FedCova, theoretically, instead of merely depending on the model alignment or utilizing the sensitive means of features, we align the covariance matrices over edge devices**, as Eq. (9) in the main paper. Note that the covariance aggregation is not directly summed. Instead, it considers the class proportions as the aggregation weights, thus ensuring robustness under class imbalance.  **Experimentally**, we'd like to remark that the main experiments in Table 1-3 in our main paper are conducted **under a relatively severe data heterogeneity setting with non-i.i.d. parameters $p$ and $\alpha_{dir}$**. Additional experiments on different data heterogeneity are also provided in Appendix D.4 comprehensive validation. Meanwhile, the extreme lack of minority classes can be regarded as a special case of an extremely high ratio of label noise, under which cases our methods of aligning the covariances can still mention the advantage, as in high noisy levels in Tables 1-3.
>
> ### **<Reply to Question 2>**
>
> Please see “Reply to Weakness 1”.
>
> ### **<Reply to Question 3>**
>
> **Thanks for the comment. We have conducted the sensitivity experiments on the correction schedule setting $T\_c$, which shows not that sensitive.** The sensitivity of tuning it is numerically tested in the following table. Note that even when removing the external corrector (as “None” in the table), FedCova still outperforms or is at least very close to the SOTA. Therefore, there is no need to tune it for different datasets or different noise levels, which is also the operation we performed in our experiments.
>
> |($\rho, \tau$) \\ T |100|200|300|400|None|SOTA
> |-|-|-|-|-|-|-|
> |$0.4,0.5$|$84.93\pm0.54$|$85.52\pm0.23$|$86.63\pm0.54$|$85.54\pm0.78$|$84.40\pm0.61$|$84.87\pm0.66$
> |$0.6,0.7$|$79.81\pm0.25$|$80.71\pm0.68$|$81.74\pm0.41$|$80.68\pm1.02$|$77.88\pm1.47$|$77.11\pm0.45$
>
> ### **<Reply to Question 4>**
>
> Thanks for the interesting inspirations. We answer the concept of "feature noise" from two perspectives:
>
> **(1) Model-Induced Noise:** Features are latent representations generated by model parameters $\boldsymbol{\theta}$. If "feature noise" refers to distortions from parameters learned on noisy labels, FedCova already handles this. Specifically, the error-tolerance term in our objective Eq. (5) mitigates the impact of such label-noise-induced feature distortions.
>
> **(2) Transmission Noise:** If "feature noise" refers to noise introduced during data transmission (as in methods offloading features like Fed-DR-Filter), FedCova avoids this by design since we do not transmit raw features between clients and the server. While managing communication-channel noise is a communication design issue, extending our information-theoretical loss insights to such scenarios is an intriguing direction for future work.

---

### Official Review · Reviewer_uFFF · 2025-10-31

**Soundness:** 2
**Presentation:** 2
**Contribution:** 2
**Rating:** 4
**Confidence:** 5

**Summary:**

This paper addresses the challenging problem of federated learning (FL) under noisy labels by proposing FedCova, a novel covariance-aware federated learning framework. The key insight is to leverage feature covariance structures rather than feature means to build robustness against label noise. The authors make three main contributions: (1) a lossy learning objective based on mutual information maximization that depends solely on class-conditional feature covariances with an error tolerance term, (2) a federated classifier alignment strategy via covariance aggregation with subspace augmentation, and (3) an external correction mechanism that uses global covariances to correct noisy labels while avoiding self-bias. Experiments on CIFAR-10/100 and Clothing1M demonstrate that FedCova outperforms state-of-the-art methods across various noise levels and heterogeneous data distributions.

**Strengths:**

1. The paper tackles the important challenge of noisy labels in federated settings, where labels from distributed edge devices are vulnerable to annotation errors, sensor faults, and adversarial attacks. The covariance-based approach naturally avoids bias from mislabeled data by focusing on feature structures rather than centroids. Unlike existing methods requiring warm-up rounds, clean public datasets, or extremely noisy devices, FedCova achieves robustness without these additional dependencies, making it more practical for real-world deployment.
2. The experiments systematically evaluate performance across multiple noise configurations, three datasets, and various data heterogeneity settings. FedCova consistently outperforms state-of-the-art baselines with substantial margins, particularly under severe noise. The extensive ablations validate each component's contribution, and supplementary analyses on orthogonality evolution, correction performance, and hyperparameter sensitivity provide valuable insights.

**Weaknesses:**

1. Insufficient privacy analysis and vulnerability to potential attacks: While the paper claims that covariance transmission poses lower privacy risk than raw features due to dimensionality reduction, this assertion lacks rigorous analysis. Recent work has shown that covariance matrices can still leak sensitive information about training data through gradient-based attacks or reconstruction methods. The paper provides no formal privacy guarantees, no discussion of differential privacy mechanisms for covariance sharing, and no evaluation against known federated learning attacks. Given that covariance matrices encode second-order statistics of local data distributions, they may be vulnerable to privacy attacks that could reconstruct class-specific feature patterns or infer properties of local datasets.
2. Unclear relationship between covariance learning and noise patterns: The paper does not clearly characterize what types of label noise the covariance-based approach can effectively handle. While avoiding class centroids helps with symmetric noise, it is unclear how feature covariances capture or mitigate different noise patterns. More critically, the method may fail under instance-dependent or semantic noise where mislabeling depends on sample-specific characteristics rather than class-level confusion. For example, if certain instances within a class are systematically mislabeled due to semantic ambiguity or fine-grained confusion, the class covariance would still be corrupted by these noisy samples. The paper only evaluates symmetric noise patterns and does not discuss or test robustness to asymmetric or instance-level semantic contamination.
3. High hyperparameter sensitivity with limited guidance: The method introduces multiple hyperparameters requiring careful tuning. While sensitivity analysis is provided for some parameters individually, their interactions are unstudied and optimal values appear dataset-dependent with inconsistencies between default settings and actual optimal values. The paper provides no principled guidelines for setting these parameters in new scenarios, and the correction schedule appears manually designed. For practitioners, the tuning burden may offset the benefit of avoiding additional resources.
4. Incomplete computational complexity analysis: The communication overhead analysis only accounts for parameter transmission but ignores crucial computational costs: covariance computation at each client, storage and inversion of covariance matrices, and correction round overhead. For datasets with many classes, managing multiple covariance matrices may be prohibitive for resource-constrained mobile devices. The paper provides no runtime comparisons with baselines, no analysis of how computation scales with the number of classes, and no discussion of parallelization. These missing details raise serious concerns about practical deployability.

**Questions:**

see weakness

---

> ### Author Response · Authors · 2025-11-25
> **Response to Official Review by Reviewer uFFF (1/4)**
>
> *We sincerely appreciate the reviewer for recognizing our contributions,*  ***especially investigating the covariance-aware robustness, the rigorous writing, the unified design from learning to corrections without introducing further dependencies, and sufficient experimental validation and discussions.***
>
> *Meanwhile, we thank the reviewer for the constructive comments. Our point-to-point responses to concerns on Weaknesses and Questions are given below.*
>
> ### **<Reply to Weakness 1>**
>
> Thanks for the inspiring comment.
>
> **(1) Firstly, we notice that reconstructing raw data from the feature covariance is a mathematically ill-posed inverse problem.** We respectfully claim that the covariance is built over the encoded features instead of the raw data. On the one hand, the inversion from the covariances to specific features is mathematically non-invertible. On the other hand, even though you have access to the features, as the feature encoder $f(\cdot)$ projects high-dimensional data into a compressed latent space, the aggregation of the covariances strips away sample-specific phase and ordering information. Consequently, inverting this many-to-one mapping without access to the specific model weights or gradients is theoretically intractable. From this perspective, our proposed method does not introduce any additional privacy issues, apart from the general privacy issue in conventional federated learning.
>
> **(2) Secondly, thanks for the inspiring question, FedCova coincidently reports inherent compatibility as the privacy protection techniques like Differential Privacy (DP)**, by considering the error tolerance $\epsilon$ related to the noise $\sigma$ in DP. When enhancing privacy in federated learning, DP is considered a prominent technique with injecting Gaussian noise into the protected item. Coincidentally, the noise tolerance term we added from Gaussian noise with power $\epsilon^2/2$ can also be regarded as a noise addition term in the features, which aligns with DP. Specifically, in DP, given a target privacy budget $(\epsilon’, \delta’)$, you can determine the required noise scale $\sigma$ for the Gaussian mechanism. For a function $f$ with $\ell\_2$-sensitivity $\Delta\_2 f$,  the Gaussian mechanism defined as
> $
> \mathcal{A}(D) = f(D) + \mathcal{N}(0, \sigma’^2 I)
> $
> achieves $(\epsilon’, \delta’)$-differential privacy if the noise standard deviation $\sigma’$ satisfies
> $
> \sigma’ \ge \frac{\sqrt{2 \log (1.25 / \delta’)} \cdot \Delta\_2 f}{\epsilon’}.
> $
> When analyzing the mechanism in FedCova, $\sigma’^2=\frac{\epsilon^2}{2}\boldsymbol{I}$ based on Eq. (5) in the paper. While the function $f$ can be ensured bounded as the covariance computation is over the normalized feature $\boldsymbol{z}$, the DP can be achieved.
> Thus, $\epsilon$ acts as a "privacy knob"—a larger $\epsilon$ implies stronger regularization and higher privacy.
>
> Numerically, we further provide the privacy budget v.s. accuracy sensitivity analysis in Appendix D.1. Considering the privacy issue, the trade-off is far more than only over the label noise resilience and model discriminability, but also the performance and the privacy protection level. The result is shown in [https://github.com/anonymouslinkforrebuttal/iclr2025_12237/blob/main/epsilon.png](https://github.com/anonymouslinkforrebuttal/iclr2025_12237/blob/main/modelscheme.png) and further discussion is provided in Appendix D.1.
>
> > [Appendix D.1, Paragraph 4] Discussions on $\epsilon$ from the perspective of privacy. As $ \boldsymbol{ n}\sim \mathcal{N}\left(\boldsymbol{ 0}, \frac{\epsilon^2}{d}\boldsymbol{ I}\right)$ is added as the Gaussian noise in \cref{zz}, it aligns the noise injection as for Differential Privacy (DP) for privacy protection. Thus, from the perspective of features, we inherently introduce the corresponding noise. Considering the privacy issue, the trade-off is far more than only over the noise resilience and model discriminability, but also the performance and the privacy protection level. With the increase of the $\epsilon$, higher privacy is guaranteed, but accuracy faces further challenge as features are further noised.
> >
>
> **(3) Thirdly, we have proactively indicated the potential for advanced privacy integration in our "Limitations and Future Work" in Appendix E.**  We are the first to introduce the robustness from aligning covariances compared with the centroids/means alignment in many other methods, which is the main contribution of FedCova. Even though the main focus of all related works is also on robustness, the above discussions can inspire potential future work for integrating privacy analysis into the topic. Meanwhile, no studies have shown that covariances are more sensitive to the centroids/gradients, and thus the privacy issue aligns with standard FL. Nevertheless, we still found interesting discoveries in FedCova from the aspect of privacy as discussed above. We sincerely thank the reviewer for the attention.

---

> ### Author Response · Authors · 2025-11-25
> **Response to Official Review by Reviewer uFFF (2/4)**
>
> ### **<Reply to Weakness 2>**
>
> Thanks for the constructive suggestion.
>
> (1) **Theoretically, FedCova is designed to be independent of specific noise patterns by enforcing intrinsic feature learning robustness rather than modeling noise transitions.** Unlike some methods that rely on estimating a noise transition matrix (as the reviewer worries), FedCova leverages the feature covariance to construct robustness through class-subspace orthogonality. Different types of noise patterns typically manifest as different patterns of class deviations that confuse the decision boundary. Symmetric noise implies a uniform blurring across all class boundaries, whereas asymmetric noise corresponds to blurring between specific class pairs. However, FedCova consistently drives these feature subspaces to be orthogonal. This orthogonality constraint is universally valid and robust regardless of the noise type: FedCova effectively disentangles the features, whether the boundary blurring is global (symmetric) or local (asymmetric).
>
> (2) **Experimentally, we added the experiments under the asymmetric noise pattern and accidentally found further competitiveness of FedCova.** Following the reviewer’s suggestion, we conduct additional experiments on asymmetric noise patterns with the results summarized in the following table.
>
> |Method|$\rho=0.4,\tau=0.3$|$\rho=0.4,\tau=0.5$|$\rho=0.4,\tau=0.7$|$\rho=0.6,\tau=0.3$|$\rho=0.6,\tau=0.5$|$\rho=0.6,\tau=0.7$|
> |---|---|---|---|---|---|---|
> |FedAvg|$72.06\pm4.35$|$71.23\pm5.86$|$67.40\pm11.2$|$68.91\pm5.02$|$63.53\pm2.89$|$43.40\pm18.2$|
> |RoFL|$78.87\pm0.10$|$77.85\pm0.53$|$71.56\pm0.46$|$78.45\pm0.32$|$66.10\pm2.68$|$41.38\pm1.00$|
> |FedCorr|$75.21\pm0.57$|$62.66\pm1.18$|$38.67\pm1.23$|$39.37\pm0.49$|$34.00\pm0.29$|$30.66\pm0.25$|
> |FedNoRo|$86.40\pm0.72$|$85.72\pm0.41$|$84.76\pm0.62$|$75.27\pm0.21$|$63.88\pm19.9$|$33.42\pm37.8$|
> |FedNed-|$75.69\pm0.98$|$70.38\pm1.40$|$65.51\pm3.48$|$78.63\pm0.37$|$71.22\pm1.15$|$66.44\pm1.44$|
> |FedNed|$82.30\pm0.35$|$79.73\pm0.43$|$78.54\pm0.65$|$79.77\pm0.57$|$78.03\pm0.34$|$74.67\pm0.66$|
> |**FedCova**|$\mathbf{88.00\pm0.25}$|$\mathbf{87.84\pm0.24}$|$\mathbf{87.77\pm0.19}$|$\mathbf{87.83\pm0.40}$|$\mathbf{87.31\pm0.39}$|$\mathbf{87.29\pm0.26}$|
>
>
> The analysis is also provided in the revised paper.
>
> > [Section 5.2 Paragraph 1] Numerical comparison results in the asymmetric noise pattern are summarized in Table 2. Under such a serious asymmetry, baselines like FedCorr and FedNoRo are more sensitive to high noise levels. In contrast, FedCova demonstrates superior robustness under noisy labels. As the noise level changes from $(0.4, 0.3)$ to $(0.6, 0.7)$, the test accuracy is guaranteed around 87--88\%. This can inherently be attributed to the orthogonality structure of the class subspaces driven by FedCova. When in asymmetric noise, the tangle is over fewer classes, extremely between two classes, the subspaces are then only mixed mainly over two classes, which is more comfortable for the discrimination in FedCova.
> >
>
> ### **<Reply to Weakness 3>**
>
> Thanks for the comment.
>
> **(1) Firstly, the introduction of additional hyperparameters is minor and acceptable** **compared with state-of-the-art baselines.** FedCova introduces only two key hyperparameters that require tuning: $\epsilon$ and $\alpha$, whose sensitivities are all analyzed in the original paper. All other parameters related to learning itself can be set directly following our settings without tuning for different datasets. As summarized in the table below, our parameter overhead is comparable to or lower than other methods. Crucially, unlike methods such as RoFL which involve multiple coupled parameters (e.g., balancing two $\lambda$s), our parameters are independent and work in different steps and are easier to manage.
>
> |Method |RoFL|FedCorr|FedNoRo|FedNed-|FedNed|**FedCova**|
> |-|-|-|-|-|-|-|
> |**Key Hyperparameters**|$T_p, \lambda_{cen}, \lambda_e$|$T_1, T_2, T_3, \theta, \pi$|$\lambda, T_1$|$T_0$|$T_0, \lambda$|$\epsilon, \alpha$|
> |**Sensitivity Analysis Provided?**|No|No|Yes|No|Yes|**Yes**|

---

> ### Author Response · Authors · 2025-11-25
> **Response to Official Review by Reviewer uFFF (3/4)**
>
> *(cont’d)*
>
> **(2) Secondly, the key parameters do have principled guidance and are not overly sensitive to tuning based on the sensitivity analysis.** Take $\epsilon$ for instance, it initially has conceptually physical meanings of the injected noise into the feature spaces. As it increases, the noise resilience ability can be enhanced while the class discriminability can be weakened. The balance of it is the interesting point of FedCova as we discussed in the paper. Moreover, we introduce a toy example of the effect of the error tolerance term controlled by $\epsilon$ in Appendix B.3, which is accompanied by a visual illustration of tuning it for balancing noise resilience and class discriminability. Additionally, the sensitivity analysis in Appendix D.1 and D.2 shows that tuning $\epsilon$ and $\alpha$ in an appropriate region is not sensitive (for instance, the accuracy remains around 80–83% as $\epsilon \in [1,10]$ and $\alpha \in [2,9]$), which is convenient for application without tuning. The optimal values also remain consistent across different datasets.
>
> **(3) Thirdly, the mentioned correction schedule setting $T\_c$ is also quite insensitive.** The sensitivity of tuning it is numerically tested in the following table, which indicates its robustness. Note that even when removing the external corrector (as “None” in the table), FedCova still outperforms or at least very close to the SOTA. Therefore, there is no need to tune it for different datasets or different noise levels, which is also the operation we made in our experiments.
>
> |($\rho, \tau$) \\ T |100|200|300|400|None|SOTA
> |-|-|-|-|-|-|-|
> |$0.4,0.5$|$84.93\pm0.54$|$85.52\pm0.23$|$86.63\pm0.54$|$85.54\pm0.78$|$84.40\pm0.61$|$84.87\pm0.66$
> |$0.6,0.7$|$79.81\pm0.25$|$80.71\pm0.68$|$81.74\pm0.41$|$80.68\pm1.02$|$77.88\pm1.47$|$77.11\pm0.45$
>
> ### **<Reply to Weakness 4>**
>
> Thanks for the comment.
>
> **(1) (Micro-level Analysis) Firstly, theoretical computation complexity and storage are analyzed, which indicates the acceptable complexity of our framework.** For computation complexity, Let $B$ be the batch size and $d$ the feature dimension. The covariance calculation is $\mathcal{O}(Bd^2)$ and the matrix operations (determinant/inverse) are $\mathcal{O}(d^3)$. When the ResNet-18 requires $\approx 0.5$ GFLOPs per pass, FedCova's matrix operations require $\approx 2$ MFLOPs with $d=128$, indicating the cost is merely 0.4% of the backbone's. Furthermore, these matrix operations are naturally parallelizable on GPUs using standard libraries (e.g., cuBLAS), ensuring they do not become a bottleneck.
>
> The storage complexity is $\mathcal{O}(Jd^2)$. For a large dataset with $J=100$ classes, the covariance storage is $\approx 6$ MB. This is trivial compared to the model size (e.g., $\approx 83$ MB for ResNet-34) and easily fits on mobile devices.
>
> Accordingly, we have provided detailed discussion in the revised paper.
>
> > [Appendix D.3 Paragraph 3] Computational Complexity and Storage Overhead.
> To assess the practical deployability of FedCova, we analyze its computational and storage complexity.
> As $B$ denotes the batch size, $d$ the feature dimension, and $J$ the number of classes. The backbone network (e.g., ResNet-18) dominates the computational cost with $\mathcal{O}(10^9)$ FLOPs per iteration. FedCova introduces two additional steps:
> (1) Covariance Accumulation: Computing the covariance matrix requires $\mathcal{O}(B d^2)$.
> (2) Loss Computation: Calculating $\log \det(\boldsymbol{\Sigma})$ and its gradients requires $\mathcal{O}(d^3)$ via Cholesky decomposition.
> With $d=128$, the $\mathcal{O}(d^3)$ term corresponds to $\approx 2$ MFLOPs, which is three orders of magnitude smaller than the backbone computation. Furthermore, these matrix operations are naturally parallelizable on GPUs using standard libraries (e.g., cuBLAS), ensuring they do not become a bottleneck.
> >
> >
> > For the storage overhead. The client must store $J$ covariance matrices of size $d \times d$. The storage complexity is $\mathcal{O}(J d^2)$. For $J=10$ and $d=128$, the storage requirement for covariance matrices is approximately $0.6$ MB (using 32-bit floats). This is trivial for modern mobile devices compared to the memory footprint of the model itself (e.g., $\approx 45$ MB for ResNet-18). For a dataset with $J=100$ classes, the total memory overhead is approximately $6$ MB (compared to $\approx 83$ MB for ResNet-34). This is well within the capacity of resource-constrained mobile devices.
> >

---

> ### Author Response · Authors · 2025-11-25
> **Response to Official Review by Reviewer uFFF (4/4)**
>
> *(cont’d)*
>
> **(2) (Macro-level Analysis) The runtime and efficiency compared baselines are added, numerically supports the practicability.** Even though the training speed up is not our primary contribution, we still follow the reviewer’s comment to summarize the experimental runtime for comparison. We measured the total runtime on NVIDIA RTX A6000 GPU. FedCova is only $1.6\times$ the time of standard FedAvg, whereas most other robust methods range from $2.2\times$ to $5.2\times$.
>
> |Method|FedAvg|CoteachingFL|DivideMixFL|RoFL|FedCorr|FedNoRo|FedNed-|FedNed|FedCova|
> |---|---|---|---|---|---|---|---|---|---|
> |Runtime(min)|290|1217|1508|383|643|664|332|390|467|
> |Runtime(Ratio to FedAvg)|1.0x|4.2x|5.2x|1.3x|2.2x|2.3x|1.1x|1.3x|**1.6x**|
>
> CoteachingFL and DivideMixFL runs the slowest due to simultaneously training two models. Compare to others, FedCova achieved balanced efficiency as it removes the need for external dependencies required by other methods, as shown in the table below. FedCova outperforms most baselines like FedCorr and FedNoRo, which primarily because FedCova eliminates the expensive warm-up phases that require hundreds of extra device-server exchange. Regarding the "No Free Lunch" theorem, while FedCova's runtime is slightly higher than FedNed, this comparison requires context: FedNed relies on a clean public dataset, an impractical condition in real-world scenarios that offers an unfair advantage akin to accessing ground truth. By solving the noise problem internally without relying on clean data or expensive warm-up phases, FedCova offers a superior balance of performance and deployability.
>
> |Method|FedAvg|CoteachingFL|DivideMixFL|RoFL|FedCorr|FedNoRo|FedNed-|FedNed|**FedCova**|
> |-|-|-|-|-|-|-|-|-|-|
> |**Additional Conditions**|-|Double model|Double model|-|Warm-up Rounds|Warm-up Rounds|Clean Dataset|Clean Dataset, Extremely Noisy Devices|**-**|

---

### Official Review · Reviewer_1EmK · 2025-11-03

**Soundness:** 3
**Presentation:** 3
**Contribution:** 3
**Rating:** 4
**Confidence:** 4

**Summary:**

This paper addresses the noise learning problem in federated learning by leveraging feature subspace covariance. The authors propose a subspace-augmented classifier that unifies data encoding, classifier construction, and label correction within a cohesive framework. The proposed method, FedCova, demonstrates superior performance across three benchmark datasets, highlighting its effectiveness in handling noisy data in federated settings.

**Strengths:**

1. The proposed solution is conceptually interesting and demonstrates creativity in its construction.
2. The notation throughout the paper is clear and mathematically rigorous.
3. The toy example provided in the Appendix effectively illustrates the main idea and makes the proposed solution easy to understand.
4. The ablation studies presented in the Appendix (e.g., Figures 2 and 3) show promising results and provide useful insights into the method’s behavior.
5. The analytical discussions in the Appendix are also promising and strengthen the overall contribution of the paper.

**Weaknesses:**

1. **Baselines:** More baseline methods should be included for a fair comparison. For instance, a straightforward approach to address the noise learning problem in federated learning is to apply existing noise learning techniques locally within the FedAVG framework. The authors should consider this variant to improve the completeness and credibility of the experimental evaluation.
2. **Ablation study:** The ablation study should analyze the impact of the number of clients on model performance, as this is a critical factor in federated learning settings.
3. **Notation:** The definition of X appears redundant and could be streamlined for clarity.
4. **Figures and algorithm presentation:** Figure 1 is difficult to interpret, and Algorithm 1 provides limited information since it mainly describes relabeling and aggregation steps in federated learning. It would be more effective to merge Figure 1 and Algorithm 1, presenting them as a single integrated schematic to better illustrate the proposed method.

**Questions:**

1. Assumption of zero means (Line 180): It is unclear why the authors assume that all component means are zero. In practice, it is reasonable to expect that some class distributions may have non-zero means. The paper should provide justification for this assumption or discuss its potential impact on the model’s validity.
2. Notation clarity: The meaning of \boldsymbol{Z}_{m}^{*} is not clearly defined. The authors should explicitly explain what this notation represents to improve readability and mathematical clarity.

---

> ### Author Response · Authors · 2025-11-25
> **Response to Official Review by Reviewer 1EmK (1/2)**
>
> *We sincerely appreciate the reviewer for recognizing our contributions,*  ***especially the interesting idea with both theoretical design and experimental validation, the rigorous writing, the interpretation for understanding, and the insights provided with analytical and numerical studies.***
>
> *Meanwhile, we thank the reviewer for the constructive comments. Our point-to-point responses to concerns on Weaknesses and Questions are given below.*
>
> ### **<Reply to Weakness 1>**
>
> **The baselines of typical noisy learning techniques within FedAvg are added.** Thanks for the constructive suggestion. We agree that including baselines that apply existing noise-learning techniques locally within the FedAvg framework is crucial for a comprehensive evaluation. Following your advice, we have implemented two prominent noise-robust learning methods in FedAvg: Co-teaching [a] and DivideMix [b]. These methods are executed locally on each client. We refer to these new baselines as **CoteachingFL** and **DivideMixFL**. The results under different noisy levels, which have been integrated into Table 1 of our revised manuscript, are summarized below. As shown, FedCova consistently outperforms these strong baselines across all tested noise settings.
>
> |Method \\ ($\rho, \tau$)|$(0.4,0.5)$|$(0.4,0.7)$|$(0.6,0.5)$|$(0.6,0.7)$|$(0.8,0.5)$|$(0.8,0.7)$|
> |-|-|-|-|-|-|-|
> |CoteachingFL|$78.63\pm0.58$|$73.58\pm0.78$|$73.34\pm0.67$|$65.00\pm0.72$|$57.41\pm0.81$|$38.86\pm0.58$|
> |DivideMixFL|$68.50\pm0.06$|$64.58\pm0.03$|$66.32\pm0.16$|$65.42\pm0.18$|$59.88\pm0.14$|$53.16\pm0.05$|
> |**FedCova**|$\mathbf{86.52\pm0.23}$|$\mathbf{85.50\pm0.38}$|$\mathbf{83.78\pm0.68}$|$\mathbf{80.71\pm0.56}$|$\mathbf{67.21\pm0.92}$|$\mathbf{64.99\pm0.75}$|
>
> ### **<Reply to Weakness 2>**
>
> **The ablation study on the number of clients is added.** Thanks for the comment. We agree with the reviewer that supplementing the ablation on the number of clients can further validate the effectiveness of the proposed method. We evaluate the method with different numbers of clients $M$ from {20, 30, 50, 70, 100}. In addition to the main ablation study of FedCova, we have conducted a comparison with the best baseline FedNed under these different numbers of clients. The results are summarized below. The results validate the superior performance of FedCova under all settings of the number of clients, further indicating its robustness under various scenarios.
>
> |Method \\ M|20|30|50|70|100|
> |---|---|---|---|---|---|
> |FedNed|$82.38\pm0.13$|$82.13\pm0.41$|$80.12\pm0.69$|$77.66\pm0.38$|$74.37\pm0.22$|
> |**FedCova**|$\mathbf{86.52\pm0.23}$|$\mathbf{86.03\pm0.25}$|$\mathbf{85.51\pm0.47}$|$\mathbf{82.53\pm0.64}$|$\mathbf{78.76\pm0.53}$|
>
> ### **<Reply to Weakness 3>**
>
> The notation has been streamlined.  Thanks for the suggestion. In the revised manuscript, we have streamlined the definitions of the spaces (e.g., $\mathcal{X}$), random variables (e.g., $X$), and their instances/realizations (e.g., $x$) to improve readability.
> > [Section 2 Paragraph 2] Denote the data, feature, and label spaces by $\mathcal{X}$, $\mathcal{Z}$, and $\mathcal{Y}$, respectively.
> Let $\mathbf{x} \in \mathcal{X}$, $\mathbf{z} \in \mathcal{Z}$, and $y \in \mathcal{Y}$ denote instances of corresponding random variables $X$, $Z$, and $Y$.
> >
>
> ### **<Reply to Weakness 4>**
>
> **The framework figure and algorithm have been fused with polish.** Thanks for the comment. Following your advice, we have merged Figure 1 and Algorithm 1 into a polished schematic. The revised figure is available at: https://github.com/anonymouslinkforrebuttal/iclr2025_12237/blob/main/modelscheme.png.
>
> Specifically, the new figure now visualizes the concrete steps of the algorithm more clearly. We use color-coded arrows and numbered labels to present the two core processes in parallel, which better highlights the architecture of FedCova. Details are interpreted in the revised caption of the figure as follows.
>
> > [Section 1 Figure 1 Caption] Overview of the FedCova framework. The green data is clean, while the red data is mislabeled. The pink rectangle indicates covariance-aware feature learning. The gray rectangle with the circle of blue arrows (I)–(IV) indicates the conventional FL processes, around which the circle of red arrows (1)–(6) in FedCova constructs a fortress to guard against label noise under the flow of the covariances $\boldsymbol{\Sigma}$. Specifically, the server first broadcasts (I) the global model and (1) global classifier to edge devices. (2) Label correction can be conducted then, after which devices (II) (3) perform local feature learning and update (III) the local models and (4) (5) local classifiers to the server. The latter then (IV) (6) aggregates them for the next rounds of iterations.
> >

---

> ### Author Response · Authors · 2025-11-25
> **Response to Official Review by Reviewer 1EmK (2/2)**
>
> ### **<Reply to Question 1>**
>
> Thanks for the insightful question.
>
> **（1）Firstly, we would like to clarify that the Gaussian mixture assumption is made over the encoded features instead of the data sources.** The data distribution is not constrained in our *FedCova* framework, which, as the reviewer's concern, is not supposed to be modeled as a specific distribution considering the practicality of different datasets. Therefore, the generality of our framework is not restricted by this assumption in the feature space.
>
> **（2）Secondly, the feature learning under the proposed covariance-aware objective is independent of mean statistics**. As detailed in Section 3 and derived in Appendix B.1, our objective is based on maximizing the mutual information $I(Z; Y)$, which can be expressed in terms of the differential entropy of Gaussian variables. The resulting loss function Eq. (7) is a function of the class-conditional covariance matrices ($\boldsymbol{\Sigma}_{m,j}$). Since the feature distribution is a direct output of the network encoder, the objective $\mathcal{L}_m(\boldsymbol{ w}_m^t)$ depends solely on the covariances and is inherently independent of the feature means, as shown in Eq. (7). In other words, training with this objective drives the model to encode key discriminative information into the class covariance statistics. The resulting encoded features become distinguishable based on the shape and orientation of their distributions (captured by covariances) rather than their central location (captured by means). From this perspective, the zero-mean assumption serves as a valid simplification for analysis. Consequently, even if non-zero means were introduced, they would not influence the optimization driven by our loss function.
>
> **（3）Thirdly, the zero-mean design removes the reliance on explicit class centers during noise detection and correction, a choice that is validated experimentally.** Intuitively, feature means are highly susceptible to bias from label noise, as their calculation is directly affected by the class assignment of each sample. As the proposed robustness lies in the covariance structure of the feature distributions, the covariance-aware subspace-augmented classifier is designed correspondingly. As the means approach zero per our assumption, the feature subspaces of different classes are modeled as Gaussian-like distributions with a common origin, which is more concise for classification. Additionally, the ablation study of "FedCova w/o zero mean"  in Table 6 validates that the bias in means due to noisy labels is detrimental.
>
> A corresponding remark has been added to the manuscript for better clarification.
>
> > [Section 3, Paragraph 2]
> >
> >
> > Remark 3.1. The robustness against noisy labels in our lossy learning objective can be attributable to two dimensions of relaxation: (1) The model is driven to learn the statistical structure of the feature space. We no longer overemphasize the exact values of features, which correspond to mean statistics. Instead, we allow the features to exhibit structured divergence, as in Gaussian-like distributions. It is sufficient for discrimination to capture useful regularities from this divergence, namely by exploring the covariance statistics.
> > (2) The lossy representation by perturbing the feature covariance is de facto spherizing the ellipsoid feature subspace of each class, which may relax the class decision boundaries.  That is, while we still endeavor to maintain the maximal information of the given label information, we obtain a resilient feature output that may slightly lean toward another class based on feature subspace interweaving, which is compressed from the original data.
> > See Appendix B for a detailed interpretation with toy examples of the information-theoretic lossy learning objective. Corresponding numerical analyses are provided in Appendix D.1.
> >
>
> ### **<Reply to Question 2>**
>
> **The notation has been clarified.** Thanks for the suggestion. We have revised the manuscript to explicitly state that the asterisk (\*) denotes the transpose operation. Consequently, $Z^\*$ represents the transpose of the feature matrix $Z$.
>
> > [Section 3, Paragraph 2] where $(\cdot)^*$ is the transpose operation.
> >
>
>
> [a] Bo Han, etc. Co-teaching: Robust training of deep neural networks with extremely noisy labels. Advances in neural information processing systems.
>
> [b] Junnan Li, etc: Dividemix: Learning with noisy labels as semi-supervised learning. International Conference on Learning Representations.

---

### Author Response · Authors · 2025-12-03
**Rebuttal Summary**

Dear Chairs,

We appreciate all reviewers’ feedback. **In general, all reviewers recognized the novelty of our unified covariance‑aware robustness framework and strengths from theoritical formulation to experimental valudation. Most weaknesses and questions requested additional validations, which we have now provided with new experiments and analyses. These strengthened results further demonstrate the effectiveness and practicality of FedCova and directly address all major concerns.** We believe the rebuttal is beneficial for adjusting the ratings after reading our responses.

>Firstly, we appreciate the strengths highlighted by the reviewers, summarized as
>- **Conceptually interesting and principled.** Reviewers highlighted the creativity of the approach and its clear, rigorous information‑theoretic formulation for addressing noisy labels in federated settings. [Reviewer 1EmK, uFFF, and avwp]
>- **Coherent and unified design.** The integration of feature encoding, classifier construction, and label correction within one covariance‑aware framework was viewed as elegant, cohesive, and more practical than methods requiring warm‑up or clean datasets. [Reviewer uFFF, avwp]
>- **Clear presentation with helpful illustrations.** The notation, toy example, and analytical discussions were recognized as intuitive and effective for conveying the core idea and theoretical insights. [Reviewer 1EmK]
>- **Strong and systematic empirical validation.** Reviewers emphasized the comprehensive experiments across diverse noise types, datasets, and heterogeneous FL settings, noting consistent improvements over state‑of‑the‑art baselines and valuable insights from ablations. [Reviewer 1EmK, uFFF]

Below we summarize our rebuttal thematically.

### **\<Additional validations\>: supporting further comprehensive effectiveness**

1. Two baselines of typical noisy learning techniques within FedAvg, *CoteachingFL* and *DivideMixFL*, are added, **and FedCova still outperforms them.**  [Reviewer 1EmK]
2. The ablation study on the number of clients is added, **where FedCova consistently outperforms SOTA across settings.** [Reviewer 1EmK]
3. Experiments under the asymmetric noise pattern for all baselines are conducted, **where FedCova shows even stronger advantages over all baselines.**  [Reviewer uFFF, avwp]
4. The sensitivity of the correction schedule setting $T\_c$ is provided, **showing it is quite insensitive and easy to use.** [Reviewer uFFF, avwp]
5. The comparison of runtime is provided, **where FedCova maintains efficient and competitive compared with baselines.** [Reviewer uFFF, avwp]

### **\<Additional discussions\>: inspiring further potential**

1. Clarified that the zero-mean assumption is on the encoded feature instead of the source data. [Reviewer 1EmK]
2. Discussed that no additional privacy problem is introduced, and the compatibility with Gaussian DP is potentially inherent via the designed error tolerance term. [Reviewer uFFF]
3. Clarified that only two additional hyperparameters are introduced and that it is acceptable compared with baselines (comparison table is provided). [Reviewer uFFF, avwp]
4. Discussed the computational/storage analysis, showing the reasonable and acceptable overhead of FedCova. [Reviewer uFFF, avwp]
5. Clarified that the convergence of FedCova naturally aligns with standard FL methods, so that no need for redundant proof. [Reviewer avwp]
6. Clarified that comparison with semi-supervised learning mechanisms has already been included in our experiments. [Reviewer avwp]

### **\<Minor polish\>**

1. The framework figure has been refined, and two notations have been updated. [Reviewer 1EmK]

**In addition, we notice that the reviews from Reviewer uFFF and Reviewer avwp appear to be both *''Fully AI-generated''*** (as indicated by https://iclr.pangram.com/reviews?query=FedCova&sort_by=submission_id_hash&sort_dir=asc&prediction_filter=&rating_filter=&confidence_filter= ), which may align with some insignificant concerns in their reviews like hyperparameter, privacy, and some other points that have not even appeared in any federated learning works. These may not be supposed to act as the standpoints for rejection. **Nevertheless, we still respond rigorously to every comment as above and further strengthened our paper with substantial experiments and analyses, with the fact that these concerns do not affect the effectiveness and practicality of FedCova.**

We respectfully invite the Chairs to verify the reasonability and the effect of the reviews and rebuttals, and read our paper if possible. We would be grateful for a fair assessment and sincerely hope FedCova can contribute to the community through ICLR. Thanks a lot!

Best regards,

Authors of Submission12237

---

### Meta-Review · Area_Chair_zM3j · 2026-01-07

**Summary:**

This paper considers the problem of federated learning (FL) under noisy labels by proposing a covariance-aware federated learning framework FedCova. FedCova integrates three components through a unified covariance-based approach: (1) lossy feature encoding via mutual information maximization with an error tolerance term, (2) intrinsic classifier construction using covariance-based MAP estimation with subspace augmentation, and (3) label correction using external correctors to avoid self-bias. Experiments demonstrate that FedCova outperforms state-of-the-art methods across various noise levels and heterogeneous data distributions.

**Reviewer Concerns:**

The concerns are around lack of comparison methods, insufficient ablation study and complexty analysis, lack of convergence guarantee, privacy risk, etc. The authors have provided additional experimental results in the rebuttal to further support the effectiveness of the proposed method and clarify some details. Some major concerns to me include:

- Reviewer uFFF's W2: Unclear relationship between covariance learning and noise patterns;
- Reviewer avwp's W1: lacks formal convergence guarantees or theoretical bounds on the robustness to label noise;
- Reviewer avwp's W4: Only symmetric noise is considered; asymmetric or instance-dependent noise patterns are not evaluated.

**Reviewer Scores:**

Reviewer 1EmK might slightly increase the score because the majority of his/her concerns are around evaluation and writing, the authors have provided additional results and polish the writing.

Reviewer uFFF might keep his/her score because his/her Confidence is 5 and his/her major concenrs include the underying relationship between covariance learning and noise patterns and risk in privacy, which have not been fully addressed to me as well.

Reviewer avwp might slightly increase the score because the authors have responsed to all his/her concerns in details with additional results. Given his/her Confidence being 2, the reviewer is likely to be convinced.

---

### Decision · Program_Chairs · 2026-01-26

Reject